# Subgraph Federated Learning with Missing Neighbor Generation

**Ke Zhang**[1,4]**, Carl Yang**[1]***, Xiaoxiao Li**[2]**, Lichao Sun**[3]**, Siu Ming Yiu**[4]

[1]Emory University, [2]University of British Columbia, [3]Lehigh University, [4]University of Hong Kong

kzhang2@cs.hku.hk, j.carlyang@emory.edu,
xiaoxiao.li@ece.ubc.ca, lis221@lehigh.edu, smyiu@cs.hku.hk

## Abstract

Graphs have been widely used in data mining and machine learning due to their unique representation of real-world objects and their interactions. As graphs are getting bigger and bigger nowadays, it is common to see their subgraphs separately collected and stored in multiple local systems. Therefore, it is natural to consider the *subgraph federated learning* setting, where each local system holds a *small* subgraph that may be *biased* from the distribution of the whole graph. Hence, the subgraph federated learning aims to collaboratively train a powerful and generalizable graph mining model without directly sharing their graph data. In this work, towards the novel yet realistic setting of subgraph federated learning, we propose two major techniques: (1) FedSage, which trains a GraphSage model based on FedAvg to integrate node features, link structures, and task labels on multiple local subgraphs; (2) FedSage+, which trains a missing neighbor generator along FedSage to deal with missing links across local subgraphs. Empirical results on four real-world graph datasets with synthesized subgraph federated learning settings demonstrate the effectiveness and efficiency of our proposed techniques. At the same time, consistent theoretical implications are made towards their generalization ability on the global graphs.

## 1 Introduction

Graph mining leverages links among connected nodes in graphs to conduct inference. Recently, graph neural networks (GNNs) have gained applause with impressing performance and generalizability in many graph mining tasks [29, 11, 16, 20, 32]. Similar to machine learning tasks in other domains, attaining a well-performed GNN model requires its training data to not only be sufficient but also follow the similar distribution as general queries. While in reality, data owners often collect limited and biased graphs and cannot observe the global distribution. With heterogeneous subgraphs separately stored in local data owners, accomplishing a globally applicable GNN requires collaboration.

Federated learning (FL) [17, 35], targeting at training machine learning models with data distributed in multiple local systems to resolve the information-silo problem, has shown its advantage in enhancing the performance and generalizability of the collaboratively trained models without the need of sharing any actual data. For example, FL has been devised in computer vision (CV) and natural language processing (NLP) to allow the joint training of powerful and generalizable deep convolutional neural networks and language models on separately stored datasets of images and texts [19, 6, 18, 39, 13].

**Motivating Scenario.** Taking the healthcare system as an example, as shown in Fig. 1, residents of a city may go to different hospitals for various reasons. As a result, their healthcare data, such as demographics and living conditions, as well as patient interactions, such as co-staying in a sickroom and co-diagnosis of a disease, are stored only within the hospitals they visit. When any healthcare problem is to be studied in the whole city, *e.g.*, the prediction of infections when a pandemic occurs,

---

*Corresponding author.

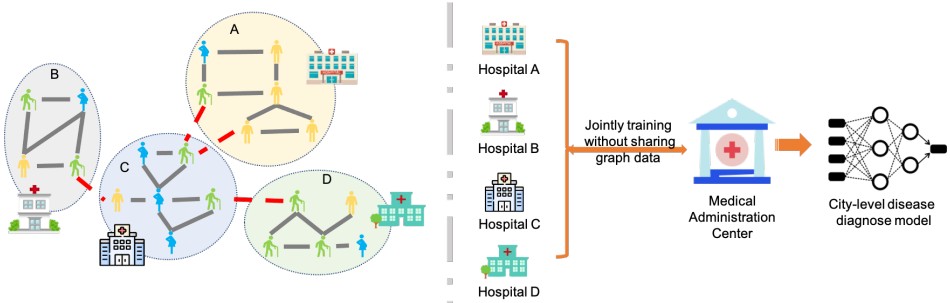

Figure 1: **A toy example of the distributed subgraph storage system:** In this example, there are four hospitals and a medical administration center. The global graph records, for a certain period, the city's patients (nodes), their information (attributes), and interactions (links). Specifically, the left part of the figure shows how the global graph is stored in each hospital, where the grey solid lines are the links explicitly stored in each hospital, and the red dashed lines are the cross-hospital links that may exist but are not stored in any hospital. The right part of the figure indicates our goal that without sharing actual data, the system obtains a globally powerful graph mining model.

a single powerful graph mining model is needed to conduct effective inference over the entire global patient network, which contains all subgraphs from different hospitals. However, it is rather difficult to let all hospitals share their patient networks with others to train the graph mining model due to conflicts of interests and privacy concerns.

In such scenarios, it is desirable to train a powerful and generalizable graph mining model over multiple distributed subgraphs without actual data sharing. However, this novel yet realistic setting brings two unique technical challenges, which have never been explored so far.

**Challenge 1: How to jointly learn from multiple local subgraphs?** In our considered scenario, the global graph is distributed into a set of small subgraphs with heterogeneous feature and structure distributions. Training a separate graph mining model on each subgraph may not capture the global data distribution and is also prone to overfitting. Moreover, it is unclear how to integrate multiple graph mining models into a universally applicable one that can handle any queries from the underlying global graph.

**Solution 1: FedSage: Training GraphSage with FedAvg.** To attain a powerful and generalizable graph mining model from small and biased subgraphs distributed in multiple local owners, we develop a framework of subgraph federated learning, specifically, with the vanilla mechanism of FedAvg [21]. As for the graph mining model, we resort to GraphSage [11], due to its advantages of inductiveness and scalability. We term this framework as FedSage.

**Challenge 2: How to deal with missing links across local subgraphs?** Unlike distributed systems in other domains such as CV and NLP, whose data samples of images and texts are isolated and independent, data samples in graphs are connected and correlated. Most importantly, in a subgraph federated learning system, data samples in each subgraph can potentially have connections to those in other subgraphs. These connections carrying important information of node neighborhoods and serving as bridges among the data owners, however, are never directly captured by any data owner.

**Solution 2: FedSage+: Generating missing neighbors along FedSage.** To deal with cross-subgraph missing links, we add a missing neighbor generator on top of FedSage and propose a novel FedSage+ model. Specifically, for each data owner, instead of training the GraphSage model on the original subgraph, it first mends the subgraph with generated cross-subgraph missing neighbors and then applies FedSage on the mended subgraph. To obtain the missing neighbor generator, each data owner impairs the subgraph by randomly holding out some nodes and related links and then trains the generator based on the held-out neighbors. Training the generator on an individual local subgraph enables it to generate potential missing links within the subgraph. Further training the generator in our subgraph FL setting allows it to generate missing neighbors across distributed subgraphs.

We conduct experiments on four real-world datasets with different numbers of data owners to better simulate the application scenarios. According to our results, both of our models outperform locally trained classifiers in all scenarios. Compared to FedSage, FedSage+ further promotes the performance of the outcome classifier. Further in-depth model analysis shows the convergence and generalization ability of our frameworks, which is corroborated by our theoretical analysis in the end.

## 2  Related works

**Graph mining.** Graph mining emerges its significance in analyzing the informative graph data, which range from social networks to gene interaction networks [31, 33, 34, 24]. One of the most frequently applied tasks on graph data is node classification. Recently, graph neural networks (GNNs), *e.g.*, graph convolutional networks (GCN) [16] and GraphSage [11], improved the state-of-the-art in node classification with their elegant yet powerful designs. However, as GNNs leverage the homophily of nodes in both node features and link structures to conduct the inference, they are vulnerable to the perturbation on graphs [4, 40, 41]. Robust GNNs, aiming at reducing the degeneration in GNNs caused by graph perturbation, are gaining attention these days. Current robust GNNs focus on the sensitivity towards modifications on node features [3, 42, 15] or adding/removing edges on the graph [37]. However, neither of these two types recapitulates the missing neighbor problem, which affects both the feature distribution and structure distribution.

To obtain a node classifier with good generalizability, the development of domain adaptive GNN sheds light on adapting a GNN model trained on the source domain to the target domain by leveraging underlying structural consistency [38, 36, 28]. However, in the distributed system we consider, data owners have subgraphs with heterogeneous feature and structure distributions. Moreover, direct information exchanges among subgraphs, such as message passing, are fully blocked due to the missing cross-subgraph links. The violation of the domain adaptive GNNs' assumptions on alignable nodes and cross-domain structural consistency denies their usage in the distributed subgraph system.

**Federated learning.** FL is proposed for cross-institutional collaborative learning without sharing raw data [17, 35, 21]. FedAvg [21] is an efficient and well-studied FL method. Similar to most FL methods, it is originally proposed for traditional machine learning problems [35] to allow collaborative training on silo data through local updating and global aggregation. The ecently proposed meta-learning framework [9, 23, 14] that exploits information from different data sources to obtain a general model attracts FL researchers [8]. However, meta-learning aims to learn general models that easily adapt to different local tasks, while we learn a generalizable model from diverse data owners to assist in solving a global task. In the distributed subgraph system, to obtain a globally applicable model without sharing local graph data, we borrow the idea of FL to collaboratively train GNNs.

**Federated graph learning.** Recent researchers have made some progress in federated graph learning. There are existing FL frameworks designed for the graph data learning task [12, 27, 30]. [12] design graph-level FL schemes with graph datasets dispersed over multiple data owners, which are inapplicable to our distributed subgraph system construction. [27] proposes an FL method for the recommendation problem with each data owner learning on a subgraph of the whole recommendation user-item graph. It considers a different scenario assuming subgraphs have overlapped items (nodes), and the user-item interactions (edges) are distributed but completely stored in the system, which ignores the possible cross-subgraph information lost in real-world scenarios. However, we study a more challenging yet realistic case in the distributed subgraph system, where cross-subgraph edges are totally missing.

In this work, we consider the commonly existing yet not studied scenario, *i.e.*, distributed subgraph system with missing cross-subgraph edges. Under this scenario, we focus on obtaining a globally applicable node classifier through FL on distributed subgraphs.

## 3  FedSage

In this section, we first illustrate the definition of the distributed subgraph system derived from real-world application scenarios. Based on this system, we then formulate our novel subgraph FL framework and a vanilla solution called FedSage.

### 3.1  Subgraphs Distributed in Local Systems

**Notation.**  We denote a global graph as $G = \{V, E, X\}$, where $V$ is the node set, $X$ is the respective node feature set, and $E$ is the edge set. In the FL system, we have the central server $S$, and $M$ data owners with distributed subgraphs. $G_i = \{V_i, E_i, X_i\}$ is the subgraph owned by $D_i$, for $i \in [M]$.

**Problem setup.**  For the whole system, we assume $V = V_1 \cup \cdots \cup V_M$. To simulate the scenario with most missing links, we assume no overlapping nodes shared across data owners, namely $V_i \cap V_j = \emptyset$ for $\forall i, j \in [M]$ and $i \neq j$. Note that the central server $S$ only maintains a graph mining model with no actual graph data stored. Any data owner $D_i$ cannot directly retrieve $u \in V_j$ from another data

owner $D_j$. Therefore, for an edge $e_{v,u} \in E$, where $v \in V_i$ and $u \in V_j$, $e_{v,u} \notin E_i \cup E_j$, that is, $e_{v,u}$ might exist in reality but is not stored anywhere in the whole system.

For the global graph $G = \{V, E, X\}$, every node $v \in V$ has its features $x_v \in X$ and one label $y_v \in Y$ for the downstream task, *e.g.*, node classification. Note that for $v \in V$, $v$'s feature $x_v \in \mathbb{R}^{d_x}$ and respective label $y_v$ is a $d_y$-dimensional one-hot vector. In a typical GNN, predicting a node's label requires an ego-graph of the queried node. For a node $v$ from graph $G$, we denote the queried ego-graph of $v$ as $G(v)$, and $(G(v), y_v) \sim \mathcal{D}_G$.

With subgraphs distributed in the system defined above, we formulate our goal as follows.

**Goal.** The system exploits an FL framework to collaboratively learn on isolated subgraphs in all data owners, without raw graph data sharing, to obtain a global node classifier $F$. The learnable weights $\phi$ in $F$ is optimized for queried ego-graphs following the distribution of ones drawn from the global graph $G$. We formalize the problem as finding $\phi^*$ that minimizes the aggregated risk

$$\phi^* = \arg\min \mathcal{R}(F(\phi)) = \frac{1}{M} \sum_i^M \mathcal{R}_i(F_i(\phi))),$$

where $\mathcal{R}_i$ is the local empirical risk defined as
$$\mathcal{R}_i(F_i(\phi)) \coloneqq \mathbb{E}_{(G_i, Y_i) \sim \mathcal{D}_{G_i}}[\ell(F_i(\phi; G_i), Y_i))],$$

where $\ell$ is a task-specific loss function
$$\ell \coloneqq \frac{1}{|V_i|} \sum_{v \in V_i} l(\phi; G_i(v), y_v).$$

## 3.2 Collaborative Learning on Isolated Subgraphs

To fulfill the system's goal illustrated above, we leverage the simple and efficient FedAvg framework [21] and fix the node classifier $F$ as a GraphSage model. The inductiveness and scalability of the GraphSage model facilitate both the training on diverse subgraphs with heterogeneous query distributions and the later inference upon the global graph. We term the GraphSage model trained with the FedAvg framework as FedSage.

For a queried node $v \in V$, a globally shared $K$-layer GraphSage classifier $F$ integrates $v$ and its $K$-hop neighborhood on graph $G$ to conduct prediction with learnable parameters $\phi = \{\phi^k\}_{k=1}^K$. Taking a subgraph $G_i$ as an example, for $v \in V_i$ with features as $h_v^0 = x_v$, at each layer $k \in [K]$, $F$ computes $v$'s representation $h_v^k$ as

$$h_v^k = \sigma\left(\phi^k \cdot \left(h_v^{k-1} || Agg\left(\left\{h_u^{k-1}, \forall u \in \mathcal{N}_{G_i}(v)\right\}\right)\right)\right), \tag{1}$$

where $\mathcal{N}_{G_i}(v)$ is the set of $v$'s neighbors on graph $G_i$, $||$ is the concatenation operation, $Agg(\cdot)$ is the aggregator (*e.g.*, mean pooling) and $\sigma$ is the activation function (*e.g.*, ReLU).

With $F$ outputting the inference label $\widetilde{y}_v = \text{Softmax}(h_v^K)$ for $v \in V_i$, the supervised loss function $l(\phi|\cdot)$ is defined as follows

$$\mathcal{L}^c = l(\phi|G_i(v), y_v) = CE(\widetilde{y}_v, y_v) = -\left[y_v \log \widetilde{y}_v + (1 - y_v) \log(1 - \widetilde{y}_v)\right], \tag{2}$$

where $CE(\cdot)$ is the cross entropy function, $G_i(v)$ is $v$'s K-hop ego-graph on $G_i$, which contains the information of $v$ and its K-hop neighbors on $G_i$.

In FedSage, the distributed subgraph system obtains a shared global node classifier $F$ parameterized by $\phi$ through $e_c$ epochs of training. During each epoch $t$, every $D_i$ first locally computes $\phi_i \leftarrow \phi - \eta \nabla \ell(\phi|\{(G_i(v), y_v)|v \in V_i^t\})$, where $V_i^t \subseteq V_i$ contains the sampled training nodes for epoch $t$, and $\eta$ is the learning rate; then the central server $S$ collects the latest $\{\phi_i|i \in [M]\}$; next, through averaging over $\{\phi_i|i \in [M]\}$, $S$ sets $\phi$ as the averaged value; finally, $S$ broadcasts $\phi$ to data owners and finishes one round of training $F$. After $e_c$ epochs, the entire system retrieves $F$ as the outcome global classifier, which is not limited to or biased towards the queries in any specific data owner.

Unlike FL on Euclidean data, nodes in the distributed subgraph system can have potential interactions with each other across subgraphs. However, as the cross-subgraph links cannot be captured by any data owner in the system, incomplete neighborhoods, compared to those on the global graph, commonly exist therein. Thus, directly aggregating incomplete queried ego-graph information through FedSage restricts the outcome $F$ from achieving the desideratum of capturing the global query distribution.

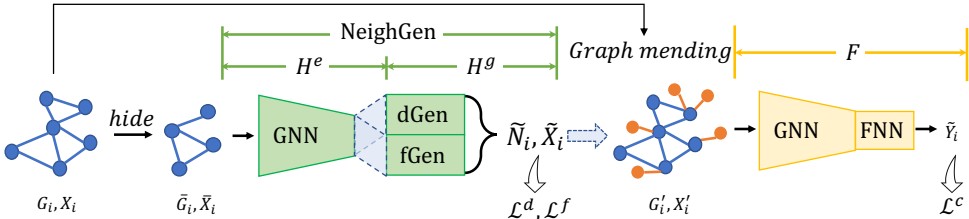

Figure 2: Joint training of missing neighbor generation and node classification.

# 4 FedSage+

In this section, we propose a novel framework of FedSage+, *i.e.*, subgraph FL with missing neighbor generation. We first design a missing neighbor generator (NeighGen) and its training schema via graph mending. Then, we describe the joint training of NeighGen and GraphSage to better achieve the goal in Section 3.1. Without loss of generality, in the following demonstration, we take NeighGen$_i$, *i.e.*, the missing neighbor generator of $D_i$, as an example, where $i \in [M]$.

## 4.1 Missing Neighbor Generator (NeighGen)

**Neural architecture of NeighGen.**    As shown in Fig. 2, NeighGen consists of two modules, *i.e.*, an encoder $H^e$ and a generator $H^g$. We describe their designs in details in the following.

$H^e$: A GNN model, *i.e.*, a K-layer GraphSage encoder, with parameters $\theta^e$. For node $v \in V_i$ on the input graph $G_i$, $H^e$ computes node embeddings $Z_i = \{z_v | z_v = h_v^K, z_v \in \mathbb{R}^{d_z}, v \in V_i\}$ according to Eq. (1) by substituting $\phi$, $G$ with $\theta^e$ and $G_i$.

$H^g$: A generative model recovering missing neighbors for the input graph based on the node embedding. $H^g$ contains dGen and fGen, where dGen is a linear regression model parameterized by $\theta^d$ that predicts the numbers of missing neighbors $\widetilde{N}_i = \{\widetilde{n}_v | \widetilde{n}_v \in \mathbb{N}, v \in V_i\}$, and fGen is a feature generator parameterized by $\theta^f$ that generates a set of $\widetilde{N}_i$ feature vectors $\widetilde{X}_i = \{\widetilde{x}_v | \widetilde{x}_v \in \mathbb{R}^{\widetilde{n}_v \times d_x}, \widetilde{n}_v \in \widetilde{N}_i, v \in V_i\}$. Both dGen and fGen are constructed as fully connected neural networks (FNNs), while fGen is further equipped with a Gaussian noise generator $\mathbf{N}(0,1)$ that generates $d_z$-dimensional noise vectors and a random sampler $R$. For node $v \in V_i$, fGen is variational, which generates the missing neighbors' features for $v$ after inserting noises into the embedding $z_v$, while $R$ ensures fGen to output the features of a specific number of neighbors by sampling $\widetilde{n}_v$ feature vectors from the feature generator's output. Mathematically, we have

$$\widetilde{n}_v = \sigma((\theta^d)^T \cdot n_v), \text{ and } \widetilde{x}_v = R\left(\sigma\left((\theta^f)^T \cdot (z_v + \mathbf{N}(0,1))\right), \widetilde{n}_v\right). \tag{3}$$

**Graph mending simulation.**    For each data owner in our system, we assume that only a particular set of nodes have cross-subgraph missing neighbors. The assumption is realistic yet non-trivial for it both seizing the quiddity of the distributed subgraph system, and allowing us to locally simulate the missing neighbor situation through a graph impairing and mending process. Specifically, to simulate a graph mending process during the training of NeighGen, in each local subgraph $G_i$, we randomly hold out $h\%$ of its nodes $V_i^h \subset V_i$ and all links involving them $E_i^h = \{e_{uv} | u \in V_i^h \text{ or } v \in V_i^h\} \subset E_i$, to form an impaired subgraph, denoted as $\bar{G}_i$. $\bar{G}_i = \{\bar{V}_i, \bar{E}_i, \bar{X}_i\}$ contains the impaired set of nodes $\bar{V}_i = V_i \setminus V_i^h$, the corresponding nodes features $\bar{X}_i = X_i \setminus X_i^h$ and edges $\bar{E}_i = E_i \setminus E_i^h$.

Accordingly, based on the ground-truth missing nodes $V_i^h$ and links $E_i^h$, the training of NeighGen on the impaired graph $\bar{G}_i$ boils down to jointly training dGen and fGen as below.

$$\mathcal{L}^n = \lambda^d \mathcal{L}^d + \lambda^f \mathcal{L}^f = \lambda^d \frac{1}{|\bar{V}_i|} \sum_{v \in \bar{V}_i} L_1^S (\widetilde{n}_v - n_v) + \lambda^f \frac{1}{|\bar{V}_i|} \sum_{v \in \bar{V}_i} \sum_{p \in [\widetilde{n}_v]} \min_{u \in \mathcal{N}_{G_i}(v) \cap V_i^h} (||\widetilde{x}_v^p - x_u||_2^2), \tag{4}$$

where $L_1^S$ is the smooth L1 distance [10] and $\widetilde{x}_v^p \in \mathbb{R}^{d_x}$ is the $p$-th predicted feature in $\widetilde{x}_v$. Note that, $\mathcal{N}_{G_i}(v) \cap V_i^h$ contains $n_v$ nodes that are $v$'s neighbors on $G_i$ missing into $V_i^h$. $\mathcal{N}_{G_i}(v) \cap V_i^h$, which can be retrieved from $V_i^h$ and $E_i^h$, provides ground-truth for training NeighGen.

**Neighbor Generation.** To retrieve $G_i'$ from $G_i$, data owner $D_i$ performs two steps, which are also shown in Fig. 2: 1) $D_i$ trains NeighGen on the impaired graph $\bar{G}_i$ w.r.t. the ground-true hidden neighbors $V_i^h$; 2) $D_i$ exploits NeighGen to generate missing neighbors for nodes on $G_i$ and then mends $G_i$ into $G_i'$ with generated neighbors. On the local graph $G_i$ alone, this process can be understood as a data augmentation that further generates potential missing neighbors within $G_i$. However, the actual goal is to allow NeighGen to generate the cross-subgraph missing neighbors, which can be achieved via training NeighGen with FL and will be discussed in Section 4.3.

### 4.2 Local Joint Training of GraphSage and NeighGen

While NeighGen is designed to recover missing neighbors, the final goal of our system is to train a node classifier. Therefore, we design the joint training of GraphSage and NeighGen, which leverages neighbors generated by NeighGen to assist the node classification by GraphSage. We term the integration of GraphSage and NeighGen on the local graphs as LocSage+.

After NeighGen mends the graph $G_i$ into $G_i'$, the GraphSage classifier $F$ is applied on $G_i'$, according to Eq. (1) (with $G_i$ replaced by $G_i'$). Thus, the joint training of NeighGen and GraphSage is done by optimizing the following loss function

$$\mathcal{L} = \mathcal{L}^n + \lambda^c \mathcal{L}^c = \lambda^d \mathcal{L}^d + \lambda^f \mathcal{L}^f + \lambda^c \mathcal{L}^c, \tag{5}$$

where $\mathcal{L}^d$ and $\mathcal{L}^f$ are defined in Eq. (4), and $\mathcal{L}^c$ is defined in Eq. (2) (with $G_i$ substituted by $G_i'$).

The local joint training of GraphSage and NeighGen allows NeighGen to generate missing neighbors in the local graph that are helpful for the classifications made by GraphSage. However, like GraphSage, the information encoded in the local NeighGen is limited to and biased towards the local graph, which does not enable it to really generate neighbors belonging to other data owners connected by the missing cross-subgraph links. To this end, it is natural to train NeighGen with FL as well.

### 4.3 Federated Learning of GraphSage and NeighGen

Similarly to GraphSage alone, as described in Section 3.2, we can apply FedAvg to the joint training of GraphSage and NeighGen, by setting the loss function to $\mathcal{L}$ and learnable parameters to $\{\theta^e, \theta^d, \theta^f, \phi\}$. However, we observe that cooperation through directly averaging weights of NeighGen across the system can negatively affect its performance, *i.e.*, averaging the weights of a single NeighGen model does not really allow it to generate diverse neighbors from different subgraphs. Recalling our goal of constructing NeighGen, which is to facilitate the training of a centralized GraphSage classifier by generating diverse missing neighbors in each subgraph, we do not necessarily need a centralized NeighGen. Therefore, instead of training a single centralized NeighGen, we train a local NeighGen$_i$ for each data owner $D_i$. In order to allow each NeighGen$_i$ to generate diverse neighbors similar to those missed into other subgraphs $G_j, j \in [M] \setminus \{i\}$, we add a cross-subgraph feature reconstruction loss into fGen$_i$ as follows:

$$\mathcal{L}_i^f = \frac{1}{|\bar{V}_i|} \sum_{v \in \bar{V}_i} \sum_{p \in [\tilde{n}_v]} \left( \min_{u \in \mathcal{N}_{G_i}(v) \cap V_i^h} (||\tilde{x}_v^p - x_u||_2^2) + \alpha \sum_{j \in [M]/i} \min_{u \in V_j} (||H_i^g(z_v)^p - x_u||_2^2) \right), \tag{6}$$

where $u \in V_j, \forall j \in [M] \setminus \{i\}$ is picked as the closest node from $G_j$ other than $G_i$ to simulate the neighbor of $v \in \bar{V}_i$ missed into $G_j$.

As shown above, to optimize Eq. (6), $D_i$ needs to pick the closest $u$ from $G_j$. However, directly transmitting node features $X_j$ in $D_j$ to $D_i$ not only violates our subgraph FL system constraints on no direct data sharing but also is impractical in reality, as it requires each $D_i$ to hold the entire global graph's node features throughout training NeighGen$_i$. Therefore, to allow $D_i$ to update NeighGen$_i$ using Eq. (6) without direct access to $X_j$, for $v \in \bar{V}_i$, $D_j$ locally computes $\sum_{p \in [\tilde{n}_v]} \min_{u \in V_j} (||H_i^g(z_v)^p - x_u||_2^2)$ and sends the respective gradient back to $D_i$.

During this process, for $v \in \bar{V}_i$, to federated optimize Eq. (6), only $H_i^g$, $H_i^g$'s input $z_v$, and the $D_j$'s locally computed model gradients of loss term $\sum_{p \in [\tilde{n}_v]} \min_{u \in V_j} (||H_i^g(z_v)^p - x_u||_2^2)$ are transmitted among the system via the server $S$. For data owner $D_i$, the gradients received from $D_j$ are then weighted by $\alpha$ and combined with the local gradients as in Eq. (6) to update the parameters of $H_i^g$ of NeighGen$_i$ In this way, $D_i$ achieves the federate training of NeighGen$_i$ without raw graph data

sharing. Note that, due to NeighGen's architecture of a concatenation of $H^e$ and $H^g$, the locally preserved GNN $H_i^e$ can prevent other data owners from inferring $x_v$ by only seeing $z_v$. Through Eq. (6), NeighGen$_i$ is expected to perceive diverse neighborhood information from all data owners, so as to generate more realistic cross-subgraph missing neighbors. The expectedly diverse and unbiased neighbors further assist the FedSage in training a globally applicable classifier that satisfies our goal in Section 3.1.

Note that, to reduce communications and computation time incurred by Eq. (6), batch training can be applied. Appendix A shows the pseudo code of FedSage+.

## 5   Experiments

We conduct experiments on four datasets to verify the effectiveness of FedSage and FedSage+ under different testing scenarios. We further conduct case studies to visualize how FedSage and FedSage+ assist local data owners in accommodating queries from the global distribution. Finally, we also provide more in-depth studies on the effectiveness of NeighGen in Appendix D.

### 5.1   Datasets and experimental settings

We synthesize the distributed subgraph system with four widely used real-world graph datasets, *i.e.*, Cora [25], Citeseer [25], PubMed [22], and MSAcademic [26]. To synthesize the distributed subgraph system, we find hierarchical graph clusters on each dataset with the Louvain algorithm [2] and use the clustering results with 3, 5, and 10 clusters of similar sizes to obtain subgraphs for data owners. The statistics of these datasets are presented in Table 1.

Table 1: Statistics of the datasets and the synthesized distributed subgraph systems with $M = 3, 5,$ and 10. #C row shows the number of classes, $|V_i|$ and $|E_i|$ rows show the averaged numbers of nodes and links in all subgraphs, and $\Delta E$ shows the total number of missing cross-subgraph links.

| Data | Cora | | | Citeseer | | | PubMed | | | MSAcademic | | |
|---|---|---|---|---|---|---|---|---|---|---|---|---|
| #C | 7 | | | 6 | | | 3 | | | 15 | | |
| $|V|$ | 2708 | | | 3312 | | | 19717 | | | 18333 | | |
| $|E|$ | 5429 | | | 4715 | | | 44338 | | | 81894 | | |
| M | 3 | 5 | 10 | 3 | 5 | 10 | 3 | 5 | 10 | 3 | 5 | 10 |
| $|V_i|$ | 903 | 542 | 271 | 1104 | 662 | 331 | 6572 | 3943 | 1972 | 6111 | 3667 | 1833 |
| $|E_i|$ | 1675 | 968 | 450 | 1518 | 902 | 442 | 12932 | 7630 | 3789 | 23584 | 13949 | 5915 |
| $\Delta E$ | 403 | 589 | 929 | 161 | 206 | 300 | 5543 | 6189 | 6445 | 11141 | 12151 | 22743 |

We implement GraphSage with two layers using the mean aggregator [5]. The number of nodes sampled in each layer of GraphSage is 5. We use batch size 64 and set training epochs to 50. The training-validation-testing ratio is 60%-20%-20% due to limited sizes of local subgraphs. Based on our observations in hyper-parameter studies for $\alpha$ and the graph impairing ratio $h$, we set $h\% \in [3.4\%, 27.8\%]$ and $\alpha=1$. All $\lambda$s are simply set to 1. Optimization is done with Adam with a learning rate of 0.001. We implement FedSage and FedSage+ in Python and execute all experiments on a server with 8 NVIDIA GeForce GTX 1080 Ti GPUs.

Since we are the first to study the novel yet important setting of subgraph federated learning, there are no existing baselines. We conduct comprehensive ablation evaluation by comparing FedSage and FedSage+ with three models, *i.e.*, 1) GlobSage: the GraphSage model trained on the original global graph without missing links (as an upper bound for FL framework with GraphSage model alone), 2) LocSage: one GraphSage model trained solely on each subgraph, 3) LocSage+: the GraphSage plus NeighGen model jointly trained solely on each subgraph.

The metric used in our experiments is the node classification accuracy on the queries sampled from the testing nodes on the global graph. For globally shared models of GlobSage, FedSage, and FedSage+, we report the average accuracy over five random repetitions, while for locally possessed models of LocSage and LocSage+, the scores are further averaged across local models.

### 5.2   Experimental results

**Overall performance.**   We conduct comprehensive ablation experiments to verify the significant promotion brought by FedSage and FedSage+ for local owners in global node classification, as

Table 2: Node classification results on four datasets with $M = 3, 5$, and 10. Besides averaged accuracy, we also provide the corresponding std.

| | Cora | | | Citesser | | |
|---|---|---|---|---|---|---|
| Model | M=3 | M=5 | M=10 | M=3 | M=5 | M=10 |
| LocSage | 0.5762 (±0.0302) | 0.4431 (±0.0847) | 0.2798 (±0.0080) | 0.6789 (±0.054) | 0.5612 (±0.086) | 0.4240 (±0.0859) |
| LocSage+ | 0.5644 (±0.0219) | 0.4533 (±0.047) | 0.2851 (±0.0080) | 0.6848 (±0.0517) | 0.5676 (±0.0714) | 0.4323 (±0.0715) |
| FedSage | 0.8656 (±0.0043) | 0.8645 (±0.0050) | 0.8626 (±0.0103) | 0.7241 (±0.0022) | 0.7226 ±0.0066) | 0.7158 (±0.0053) |
| FedSage+ | **0.8686** (±0.0054) | **0.8648** (±0.0051) | **0.8632** (±0.0034) | **0.7454** (±0.0038) | **0.7440** (±0.0025) | **0.7392** (±0.0041) |
| GlobSage | 0.8701 (±0.0042) | | | 0.7561 (±0.0031) | | |

| | PubMed | | | MSAcademic | | |
|---|---|---|---|---|---|---|
| Model | M=3 | M=5 | M=10 | M=3 | M=5 | M=10 |
| LocSage | 0.8447 (±0.0047) | 0.8039 (±0.0337) | 0.7148 (±0.0951) | 0.8188 (±0.0331) | 0.7426 (±0.0790) | 0.5918 (±0.1005) |
| LocSage+ | 0.8481 (±0.0041) | 0.8046 (±0.0318) | 0.7039 (±0.0925) | 0.8393 (±0.0330) | 0.7480 (±0.0810) | 0.5927 (±0.1094) |
| FedSage | 0.8708 (±0.0014) | 0.8696 (±0.0035) | 0.8692 (±0.0010) | 0.9327 (±0.0005) | 0.9391 (±0.0007) | 0.9262 (±0.0009) |
| FedSage+ | **0.8775** (±0.0012) | **0.8755** (±0.0047) | **0.8749** (±0.0013) | **0.9359** (±0.0005) | **0.9414** (±0.0006) | **0.9314** (±0.0009) |
| GlobSage | 0.8776(±0.0011) | | | 0.9681(±0.0006) | | |

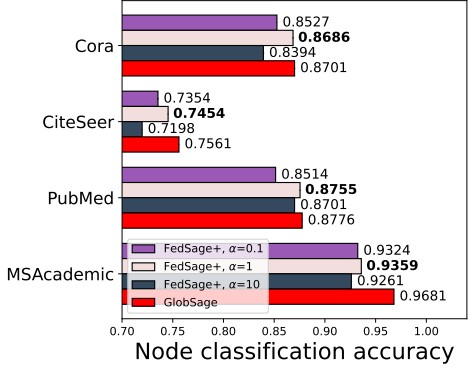

(a) Hyper-parameter study for $\alpha$ with $h = 15\%$.

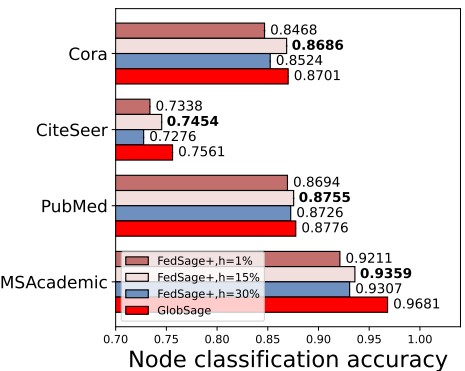

(b) Hyper-parameter study for $h$ with $\alpha = 1$.

Figure 3: Node classification results on four datasets under different $\alpha$ and $h$ values with $M=3$.

shown in Table 2. The most important observation emerging from the results is that FedSage+ not only clearly outperforms LocSage by an average of 23.18%, but also distinctly overcomes the cross-subgraph missing neighbor problem by reducing the average accuracy drop from the 2.11% of FedSage to 1.28%, when compared with GlobSage (absolute accuracy difference).

The significant gaps between a locally obtained classifier, *i.e.*, LocSage or LocSage+, and a federated trained classifier, *i.e.*, FedSage or FedSage+, assay the benefits brought by the collaboration across data owners in our distributed subgraph system. Compared to FedSage, the further elevation brought by FedSage+ corroborates the assumed degeneration brought by missing cross-subgraph links and the effectiveness of our innovatively designed NeighGen module. Notably, when the graph is relatively sparse (*e.g.*, see Citeseer in Table 1), FedSage+ significantly exhibits its robustness in resisting the cross-subgraph information loss compared to FedSage. Note that the gaps between LocSage and LocSage+ are comparatively smaller, indicating that our NeighGen serves more than a robust GNN trainer, but is rather uniquely crucial in the subgraph FL setting.

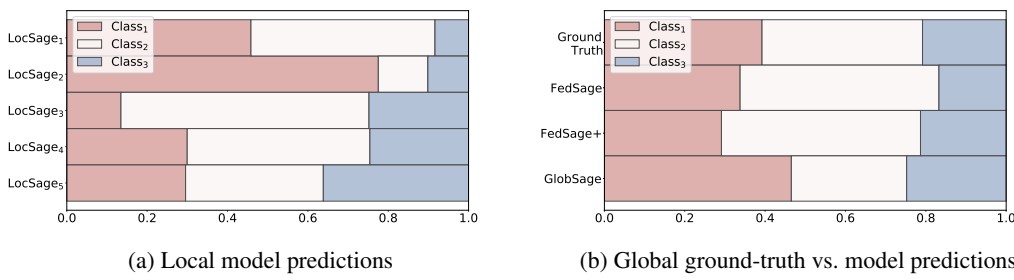

(a) Local model predictions       (b) Global ground-truth vs. model predictions

Figure 4: Label distributions on the PubMed dataset with $M$=5.

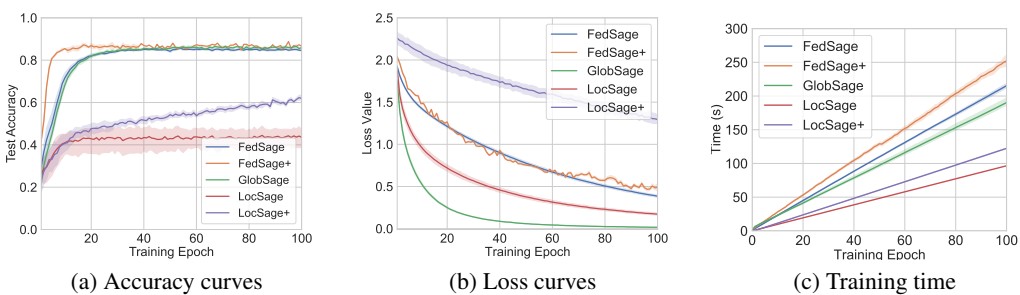

(a) Accuracy curves      (b) Loss curves      (c) Training time

Figure 5: Training curves of different frameworks (GlobSage provides an upper bound).

**Hyper-parameter studies.** We compare the downstream task performance under different $\alpha$ and $h$ values with three data owners. Results are shown in Fig. 3, where Fig. 3 (a) shows results when $h$ is fixed as 15%, and Fig. 3 (b) shows results under $\alpha$=1.

Fig. 3 (a) indicates that choosing a proper $\alpha$, which brings the information from other subgraphs in the system, can constantly elevate the final testing accuracy. Across different datasets, the optimal $\alpha$ is constantly around 1, and the performance is not influenced much unless $\alpha$ is set to extreme values like 0.1 or 10. Referring to Fig. 3 (b), we can observe that either a too-small (1%) or a too-large (30%) hiding portion can degrade the learning process. A too-small $h$ can not provide sufficient data for training NeighGen, while a too-large $h$ can result in sparse local subgraphs that harm the effective training of GraphSage. Referring back to the graph statistics in Table 1 in the paper, the portion of actual missing edges compared to the global graph is within the range of [3.4%, 27.8%], which explains why a value like 15% can mostly boost the performance of FedSage+.

**Case studies.** To further understand how FedSage and FedSage+ improve the global classifier over LocSage, we provide case study results on PubMed with five data owners in Fig. 4. For the studied scenario, each data owner only possesses about 20% of the nodes with rather biased label distributions, as shown in Fig. 4 (a). Such bias is due to the way we synthesize the distributed subgraph system with Louvain clustering, which is also realistic in real scenarios. Local bias essentially makes it hard for any local data owner with limited training samples to obtain a generalized classifier that is globally useful. Although with 13.9% of the links missing among the system, both FedSage and FedSage+ empower local data owners in predicting labels that closely follow the ground-true global label distribution as shown in Fig. 4 (b). The figure clearly evidences that our FL models exhibit their advantages in learning a more realistic label distribution as our goal in Section 3.1, which is consistent with the observed performances in Table 2 and our theoretical implications in Section 6.

For Cora dataset with five data owners, we visualize testing accuracy, loss convergence, and runtime along 100 epochs in obtaining $F$ with FedSage, FedSage+, GlobSage, LocSage and LocSage+. The results are presented in Fig. 5. Both FedSage and FedSage+ can consistently achieve convergence with rapidly improved testing accuracy. Regarding runtime, even though the classifier from FedSage+ learns from distributed mended subgraphs, FedSage+ does not consume observable more training time compared to FedSage. Due to the additional communications and computations in subgraph FL, both FedSage and FedSage+ consume slightly more training time compared to GlobSage.

# 6 Implications on Generalization Bound

In this section, we provide a theoretical implication for the generalization error associated with number of training samples, *i.e.*, nodes in the distributed subgraph system, following Graph Neural Tangent Kernel (GNTK) [7] on universal graph neural networks. Thus, we are motivated to promote the FedSage and FedSage+ algorithms that include more nodes in the global graph through collaborative training with FL.

**Setting.** Our explanation builds on a generalized setting, where we assume a GNN $F$ with layer-wise aggregation operations and fully-connected layers with ReLU activation functions, which includes GraphSage as a special case. The weights of $F$, $\phi$, is i.i.d. sampled from a multivariate Gaussian distribution $\mathbf{N}(0, I)$. For Graph $G = \{V, E, X\}$, we define the kernel matrix of two nodes $u, v \in V$ as follows. Here we consider $F$ is in the GNTK format.

**Definition 6.1 (Informal version of GNTK on node classification (Definition B.2))** *Considering in the overparameterized regime for an GNN $F$, $F$ is trained using gradient descent with infinite small learning rate. Given $n$ nodes with corresponding labels as training samples, we denote $\mathbf{\Theta} \in \mathbb{R}^{n \times n}$ as the the kernel matrix of GNTK. $\mathbf{\Theta}_{uv}$ is defined as*

$$\mathbf{\Theta}_{uv} = \mathbb{E}_{\phi \sim \mathbf{N}(0,I)} \left[ \left\langle \frac{\partial F(\phi, G, u)}{\partial \phi}, \frac{F(\phi, G, v)}{\partial \phi} \right\rangle \right] \in \mathbb{R}.$$

Full expression of $\mathbf{\Theta}$ is shown in the Appendix B. The generalization ability in the GNTK regime depends on the kernel matrix $\mathbf{\Theta}$. We present the generalization bound associated with the number of training samples $n$ in Theorem 6.2.

**Theorem 6.2 (Generalization bound)** *Given $n$ training samples of nodes $(u_i, y_i)_{i=1}^{n}$ drawn i.i.d. from the global graph $G$, consider any loss function $l : \mathbb{R} \times \mathbb{R} \mapsto [0, 1]$ that is 1-Lipschitz in the first argument such that $l(y, y) = 0$. With probability at least $1 - \sigma$ and constant $c \in (0, 1)$, the generalization error of GNTK for node classification can be upper-bounded by*

$$L_{\mathcal{D}(F)} = \mathbb{E}_{(u',y) \sim G}[l(F(G, u'), y)] \lesssim O(1/n^c).$$

Following the generalization bound analysis in [7], we use a standard generalization bound of kernel methods of [1], which shows the upper bound of our GNTK formation error depends on that of $\mathbf{y}^{\top} \mathbf{\Theta}^{(-1)} \mathbf{y}$ and $\mathrm{tr}(\mathbf{\Theta})$, where $\mathbf{y}$ is the label vector. Appendix C shows the full version of the proofs.

**Implications.** We show the error bound of GNTK on node classification corresponding to the number of training samples. Under the assumptions in Definition 6.1, our theoretical result indicates that more training samples bring down the generalization error , which provides plausible support for our goal of building a globally useful classifier through FL in Eq. (3.1). Such implications are also consistent with our experimental findings in Fig. 4 where our FedSage and FedSage+ models can learn more generalizable classifiers that follow the label distributions of the global graph through involving more training nodes across different subgraphs.

# 7 Conclusion

This work aims at obtaining a generalized node classification model in a distributed subgraph system without direct data sharing. To tackle the realistic yet unexplored issue of missing cross-subgraph links, we design a novel missing neighbor generator NeighGen with the corresponding local and federated training processes. Experimental results evidence the distinguished elevation brought by our FedSage and FedSage+ frameworks , which is consistent with our theoretical implications.

Though FedSage manifests advantageous performance, it confronts additional communication cost and potential privacy concerns. As communications are vital for federated learning, properly reducing communication and rigorously guaranteeing privacy protection in the distributed subgraph system can both be promising future directions.

## Acknowledgments and Disclosure of Funding

This work is partially supported by the internal funding and GPU servers provided by the Computer Science Department of Emory University. We thank Dr. Pan Li from Purdue University for the suggestions on the design of our NeighGen mechanism.

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
