# A  FedSage+ Algorithm

Referring to Section 4.3, FedSage+ includes two phases. Firstly, all data owners in the distributed subgraph system jointly train NeighGen models through sharing gradients. Next, after every local graph mended with synthetic neighbors generated by the respective NeighGen model, the system executes FedSage to obtain the generalized node classification model. Algorithm 1 shows the pseudo code for FedSage+.

---

**Algorithm 1** FedSage+: Subgraph federated learning with missing neighbor generation

---

1: **Notations.** Data owners set $\{D_1, \ldots, D_M\}$, server $S$, epochs for jointly training NeighGen $e_g$, epochs for FedSage $e_c$, learning rate for FedSage $\eta$.
2: For $t = 1 \rightarrow e_g$, we iteratively run **procedure A**, **procedure C**, **procedure D**, and **procedure E**
3: Every $D_i \in \{D_1, \ldots, D_M\}$ retrieves $G_i'$ from FL trained NeighGen$_i$
4: $S$ initializes and broadcasts $\phi$
5: For $t = 1 \rightarrow e_c$, we iteratively run **procedure B** and **procedure F**
6:
7: *On the server side:*
8: **procedure A** SERVEREXCECUTIONFORGEN($t$)               ▷ FL of NeighGen on epoch $t$
9:     Collect $(Z_i^t, H_i^g) \leftarrow$ LOCALREQUEST($D_i, t$) from every data owner $D_i$, where $i \in [M]$
10:     Send $\{(Z_j^t, H_j^g)|j \in [M] \setminus \{i\}\}$ to every data owner $D_i$, where $i \in [M]$
11:     **for** $D_i \in \{D_1, \ldots, D_M\}$ **in parallel do**
12:         $\{\nabla\mathcal{L}_{i,1}^f, \ldots, \nabla\mathcal{L}_{i,M}^f\} \setminus \{\nabla\mathcal{L}_{i,i}^f\} \leftarrow$ FEEDFORWARD($D_i, \{(Z_j^t, H_j^g)|j \in [M] \setminus \{i\}\}$)
13:     **for** $D_i \in \{D_1, \ldots, D_M\}$ **in parallel do**
14:         Aggregate gradients as $\nabla\mathcal{L}_{i,J}^f \leftarrow \sum_{j \in [M] \setminus \{i\}} \nabla\mathcal{L}_{i,j}^f$
15:         Send $\nabla\mathcal{L}_{i,J}^f$ to $D_i$ for UPDATENEIGHGEN($D_i, \nabla\mathcal{L}_{i,J}^f$)
16: **procedure B** SERVEREXCECUTIONFORC($t$)        ▷ FedSage for mended subgraphs on epoch $t$
17:     Collect $\phi_i \leftarrow$ LOCALUPDATEC($D_i, \phi, t$) from all data owners
18:     Broadcast $\phi \leftarrow \frac{1}{M} \sum_{i \in [M]} \phi_i$
19:
20: *On the data owners side:*
21: **procedure C** LOCALREQUEST($D_i, t$)                              ▷ Run on $D_i$
22:     Sample a $V_i^t \in \bar{V}_i^t$ and get $Z_i^t \leftarrow \{H_i^e(\bar{G}_i(v))|v \in V_i^t\}$
23:     Send $Z_i^t, H_i^g$ to Server
24: **procedure D** FEEDFORWARD($D_i, \{(Z_j^t, H_j^g)|j \in [M] \setminus \{i\}\}$)          ▷ Run on $D_i$
25:     **for** $j \in [M] \setminus \{i\}$ **do**
26:         $\mathcal{L}_{j,i}^f \leftarrow \frac{1}{|Z_i^t|} \sum_{z_v \in Z_j^t} \sum_{p \in [|H_j^g(z_v)|]} \left( \min_{u \in V_i} (||H_j^g(z_v)^p - x_u||_2^2) \right)$   ▷ A part of Eq. (6)
27:     Compute and send $\{\nabla\mathcal{L}_{1,i}^f, \ldots, \nabla\mathcal{L}_{M,i}^f\} \setminus \{\nabla\mathcal{L}_{i,i}^f\}$ to Sever
28: **procedure E** UPDATENEIGHGEN($D_i, \nabla\mathcal{L}_{i,J}^f$)                   ▷ Run on $D_i$
29:     Train NeighGen$_i$ by optimizing Eq. (6).
30: **procedure F** LOCALUPDATEC($D_i, \phi, t$)                         ▷ Run on $D_i$
31:     Sample a $V_i^t \subseteq V_i$
32:     $\phi_i \leftarrow \phi - \eta\nabla l(\phi|\{(G_i'(v), y_v)|v \in V_i^t\})$
33:     Send $\phi_i$ to Sever

---

# B  Full Version of Definition 6.1

**Notation.**   We denote the whole graph $G = \{V, E, X\}$ and $|V| = n$. To perform node classification on $G$, we consider a GNN $F$ with $K$ aggregation operations[1] and each aggregation operation contains $R$ fully-connected layers. We describe the aggregation operation below.

**Definition B.1 (Aggregation operation, (AGG))** *For $\forall k \in [K]$, AGG aggregates the information from the previous layer and performs $R$ times non-linear transformation. With denoting the initial feature vector for node $u \in V$ as $h_u^{(0,R)} = x_u \in \mathbb{R}^d$, for an AGG with $R = 2$ fully-connected layers, the AGG can be written as:*

$$h_u^{(k,0)} = c_u \sqrt{\frac{c_\sigma}{m}} \sigma \left( \phi_{k,2} \sqrt{\frac{c_\sigma}{m}} \sigma \left( \phi_{k,1} \cdot c_u \sum_{v \in \mathcal{N}(u) \cup u} h_v^{(k-1,0)} \right) \right),$$

*where $c_\sigma \in \mathbb{R}$ is a scaling factor related initialization, $c_u \in \mathbb{R}$ is a scaling factor associated with neighbor aggregation, $\sigma(\cdot)$ is ReLU activation, and learnable parameter $\phi_{k,r} \in \mathbb{R}^{m \times m}$ for $\forall (k,r) \in [K] \times [R] \backslash \{(1,1)\}$ as $\phi_{1,1} \in \mathbb{R}^{k \times d}$.*

For notation simplicity, GNN $F$ here is considered in GNTK format. The weights of $F$, $\phi$ is i.i.d. sampled from a multivariate Gaussian distribution $\mathcal{N}(0, I)$. For node $u \in V$, we denote $u$'s computational graph as $G_u = \{V_u, E_u, X_u\}$ and $|V_u| = n_u$. Let $\langle a, b \rangle$ denote inner-product of vector $a$ and $b$. We are going to define the kernel matrix of two nodes $u, v \in V$ as follows.

**Definition B.2 (GNTK for node classification)** *Considering in the overparameterized regime for an GNN $F$, $F$ is trained using gradient descent with infinite small learning rate. Given $n$ training samples of nodes with corresponding labels, we denote $\Theta \in \mathbb{R}^{n \times n}$ as the the kernel matrix of GNTK. For $\forall u, v \in V$, $\Theta_{uv}$ is the $u, v$ entry of $\Theta$ and defined as*

$$\Theta_{uv} = \mathbb{E}_{\phi \sim \mathcal{N}(0,I)} \left[ \left\langle \frac{\partial F(\phi, G, u)}{\partial \phi}, \frac{F(\phi, G, v)}{\partial \phi} \right\rangle \right]$$

$$= \mathbb{E}_{\phi \sim \mathcal{N}(0,I)} \left[ \left\langle \frac{\partial F(\phi, G_u, u)}{\partial \phi}, \frac{F(\phi, G_v, v)}{\partial \phi} \right\rangle \right] \in \mathbb{R}.$$

In GNTK formulation, an AGG B.1 needs to calculate 1) a covariance matrix $\Sigma(G_u, G_v)$; and 2) the intermediate kernel values $\Theta(G_u, G_v)$ Now, we specify the pairwise value in $\Sigma(G_u, G_v) \in \mathbb{R}^{n_u \times n_v}$ and $\Theta(G_u, G_v) \in \mathbb{R}^{n_u \times n_v}$. For $\forall k \in [K]$ and $\forall r \in [R]$, $\Sigma_{(r)}^{(k)}(G_u, G_v)$ and $\Theta_{(r)}^{(k)}(G_u, G_v)$ indicate the corresponding covariance and intermediate kernel matrix for $r$th transformation and $k$th layers. Initially, we have $[\Sigma_{(R)}^{(0)}(G_u, G_v)]_{uv} = [\Theta_{(R)}^{(0)}(G_u, G_v)]_{uv} = \langle h_u, h_v \rangle$, where $h_u, h_v \in \mathbb{R}^d$ are the input features of node $u$ and $v$. Denote the scaling factor for node $u$ as $c_u$. $\Theta_{(R)}^{(l)}(G_u, G_v)$ can be calculated recursively through the aggregation operation given in [7]. Specifically, we have the following two steps.

**Step 1: Neighborhood Aggregation**   As the AGG we defined above, in GNTK, the aggregation step can be performed as:

$$\left[ \Sigma_{(0)}^{(k)}(G_u, G_v) \right]_{uv} = c_u c_v \sum_{u' \in \mathcal{N}(u) \cup \{u\}} \sum_{v' \in \mathcal{N}(v) \cup \{v\}} \left[ \Sigma_{(R)}^{(k-1)}(G_u, G_v) \right]_{u'v'},$$

$$\left[ \Theta_{(0)}^{(k)}(G_u, G_v) \right]_{uv} = c_u c_v \sum_{u' \in \mathcal{N}(u) \cup \{u\}} \sum_{v' \in \mathcal{N}(v) \cup \{v\}} \left[ \Theta_{(R)}^{(k-1)}(G_u, G_v) \right]_{u'v'}.$$

**Step 2: $R$ transformations**   Now, we consider the $R$ ReLU fully-connected layers which perform non-linear transformations to the aggregated feature generated in step 1. The ReLU activation function

---

[1]In Graphsage, this is equivalent to having $K$ graph convectional layers.

$\sigma(x) = \max\{0, x\}$'s derivative is denoted as $\dot{\sigma}(x)$ For $r \in [R], u, v \in V$, we define covariance matrix and its derivative as

$$\left[\mathbf{\Sigma}_{(r)}^{(k)}(G_u, G_v)\right]_{uv} = c_\sigma \mathbb{E}_{(a,b)\sim\mathcal{N}\left(\mathbf{0}, \left[\mathbf{A}_{(r)}^{(k)}(G_u,G_v)\right]_{uv}\right)}[\sigma(a)\sigma(b)],$$

$$\left[\dot{\mathbf{\Sigma}}_{(r)}^{(k)}(G_u, G_v)\right]_{uv} = c_\sigma \mathbb{E}_{(a,b)\sim\mathcal{N}\left(\mathbf{0}, \left[\mathbf{A}_{(r)}^{(k)}(G_u,G_v)\right]_{uv}\right)}[\dot{\sigma}(a)\dot{\sigma}(b)],$$

where $[\mathbf{A}_{(r)}^{(k)}(G_u, G_v)]_{uv}$ is an intermediate variable that

$$\left[\mathbf{A}_{(r)}^{(k)}(G_u, G_v)\right]_{uv} = \begin{pmatrix} \left[\mathbf{\Sigma}_{(r-1)}^{(k)}(G_u, G_u)\right]_{uu} & \left[\mathbf{\Sigma}_{(r-1)}^{(k)}(G_u, G_v)\right]_{uv} \\ \left[\mathbf{\Sigma}_{(r-1)}^{(k)}(G_u, G)\right]_{uv} & \left[\mathbf{\Sigma}_{(r-1)}^{(k)}(G_v, G_v)\right]_{vv} \end{pmatrix} \in \mathbb{R}^{2\times2}$$

Thus, we have

$$\left[\mathbf{\Theta}_{(r)}^{(l)}(G_u, G_v)\right]_{uv} = \left[\mathbf{\Theta}_{(r-1)}^{(k)}(G_u, G_v)\right]_{uv}\left[\dot{\mathbf{\Sigma}}_{(r)}^{(k)}(G_u, G_v)\right]_{uv} + \left[\mathbf{\Sigma}_{(r)}^{(k)}(G_u, G_v)\right]_{uv}.$$

$\mathbf{\Theta} = \mathbf{\Theta}_{(R)}^{(l)}(G_u, G_v)$ can be viewed as kernel matrix of GNTK for node classification. The generalization ability in the NTK regime and depends on the kernel matrix.

## C Missing Proofs for Theorem 6.2

In this section, we provide the detailed version and proof of Theorem 6.2.

**Theorem C.1 (Full version of generalization bound Theorem 6.2)** *Given $n$ training data samples $(h_i, y_i)_{i=1}^n$ drawn i.i.d from Graph $G$, we consider any loss function $l : \mathbb{R} \times \mathbb{R} \mapsto [0, 1]$ that is 1-Lipschitz in the first argument such that $l(y, y) = 0$. With a probability at least $1 - \sigma$ and a constant $c \in (0, 1)$, the generalization error of GNTK for node classification can be upper-bounded by*

$$L_{\mathcal{D}(F)} = \mathbb{E}_{(G,y)\sim\mathcal{D}}[l(F(G), y)] = O\left(\frac{\sqrt{\mathbf{y}^\top \mathbf{\Theta}^{(-1)}\mathbf{y} \cdot \text{tr}(\mathbf{\Theta})}}{n} + \sqrt{\frac{\log(1/\sigma)}{n}}\right).$$

To prove the generalization bound, we make the following assumptions about the labels.

**Assumption C.2** *For each $i \in [n]$, the labels $y_i = [\mathbf{y}]_i \in \mathbb{R}$ satisfies*

$$y_i = \alpha_1 \langle \bar{h}_u^\top, \beta_1 \rangle + \sum_{l=1}^{\infty} \alpha_{2l} \langle \bar{h}_u^\top, \beta_{2l} \rangle^{2l},$$

*where $\alpha_1, \alpha_2, \cdots, \alpha_{2k} \in \mathbb{R}$, $\beta_1, \beta_2, \cdots, \beta_{2k} \in \mathbb{R}^d$, and $\bar{h}_u = c_u \sum_{v\in\mathcal{N}(u)\cup\{u\}} h_v \in \mathbb{R}^d$.*

The following Lemma C.3 and C.4 give the bounds for $\sqrt{\mathbf{y}^\top\mathbf{\Theta}^{(-1)}\mathbf{y}}$ and $\text{tr}(\mathbf{\Theta})$.

**Lemma C.3 (Bound on $\sqrt{\mathbf{y}^\top\mathbf{\Theta}^{(-1)}\mathbf{y}}$)** *Under the Assumption C.2, we have*

$$\sqrt{\mathbf{y}^\top\mathbf{\Theta}^{(-1)}\mathbf{y}} \le 2|\alpha_1|\|\beta_1\|_2 + \sum_{l=1}^{\infty}\sqrt{2\pi}(2k-1)|\alpha_{2l}|\|\beta_{2l}\|_2^{2l} = o(n)$$

*Proof.* Without loss of generality, we consider a simple GNN ($K = 1, R = 1$) in Section B and define the kernel matrix for on the computational graph $G_u, G_v$ node $u, v \in V$ as

$$\mathbf{\Theta}_{uv} = \left[\mathbf{\Sigma}_{(0)}^{(1)}(G_u, G_v)\right]_{uv}\left[\dot{\mathbf{\Sigma}}_{(1)}^{(1)}(G_u, G_v)\right]_{uv} + \left[\mathbf{\Sigma}_{(1)}^{(1)}(G_u, G_v)\right]_{uv}.$$

We decompose $\mathbf{\Theta} \in \mathbb{R}^{n \times n}$ into $\mathbf{\Theta} = \mathbf{\Theta}' + \mathbf{\Theta}''$, where

$$\mathbf{\Theta}'_{uv} = \left[\mathbf{\Sigma}^{(1)}_{(0)}(G_u, G_v)\right]_{uv} \left[\dot{\mathbf{\Sigma}}^{(1)}_{(1)}(G_u, G_v)\right]_{uv}, \text{ and } \quad \mathbf{\Theta}''_{uv} = \left[\mathbf{\Sigma}^{(1)}_{(1)}(G_u, G_v)\right]_{uv}.$$

Following the proof in [7] and assuming $\|\bar{h}_u\|_2 = 1$, we have

$$\left[\dot{\mathbf{\Sigma}}^{(1)}_{(1)}(G_u, G_v)\right]_{uv} = \frac{\pi - \arccos\left(\left[\mathbf{\Sigma}^{(1)}_{(0)}(G_u, G_v)\right]_{uv}\right)}{2\pi},$$

$$\left[\mathbf{\Sigma}^{(1)}_{(1)}(G_u, G_v)\right]_{uv} = \frac{\pi - \arccos\left(\left[\mathbf{\Sigma}^{(1)}_{(0)}(G_u, G_v)\right]_{uv}\right) + \sqrt{1 - \left[\mathbf{\Sigma}^{(1)}_{(0)}(G_u, G_v)\right]^2_{uv}}}{2\pi}.$$

Then,

$$\mathbf{\Theta}' = \frac{1}{4}\left[\mathbf{\Sigma}^{(1)}_{(0)}(G_u, G_v)\right]_{uv} + \frac{1}{2\pi}\left[\mathbf{\Sigma}^{(1)}_{(0)}(G_u, G_v)\right]_{uv} \arcsin\left(\left[\mathbf{\Sigma}^{(1)}_{(0)}(G_u, G_v)\right]_{uv}\right)$$

$$= \frac{1}{4}\left[\mathbf{\Sigma}^{(1)}_{(0)}(G_u, G_v)\right]_{uv} + \frac{1}{2\pi}\sum_{l=1}^{\infty}\frac{(2k-3)!!}{(2k-2)!! \cdot (2k-1)} \cdot \left[\mathbf{\Sigma}^{(1)}_{(0)}(G_u, G_v)\right]^{2k}_{uv}$$

$$= \frac{1}{4}\bar{h}_u^\top \bar{h}_v + \frac{1}{2\pi}\sum_{l=1}^{\infty}\frac{(2k-3)!!}{(2k-2)!! \cdot (2k-1)} \cdot \left(\bar{h}_u^\top \bar{h}_v\right)^{2k}.$$

We denote $\Phi^{2k}$ as the feature map of the kernel at degree $2k$ that $\langle h_u, h_v\rangle^{(2k)} = \Phi^{2k}(h_u)^\top \Phi^{2k}(h_v)$. Following the proof in [7], we have

$$\mathbf{\Theta}' = \frac{1}{4}\bar{h}_u^\top \bar{h}_{u'} + \frac{1}{2\pi}\sum_{l=1}^{\infty}\frac{(2k-3)!!}{(2k-2)!! \cdot (2k-1)} \cdot \Phi^{2k}(\bar{h}_u)^\top \Phi^{2k}(\bar{h}_v).$$

As $\mathbf{\Theta}''$ is a positive semidefinite matrix, we have

$$\mathbf{y}^\top \mathbf{\Theta}^{(-1)}\mathbf{y} \le \mathbf{y}^\top \mathbf{\Theta}'^{(-1)}\mathbf{y}.$$

We define $y_i^{(0)} = \alpha_1 \left(\bar{h}_u^\top\right)\beta_1$ and $y_i^{(2k)} = \alpha_{2k}\Phi^{2k}\left(\bar{h}_u\right)^\top \Phi^{2k}(\beta_{2k})$ for each $k \ge 1$. Under Assumption C.2, label $y_i$ can be rewritten as

$$y_i = y_i^{(0)} + \sum_{k=1}^{\infty} y_i^{(2k)}.$$

Then we have

$$\sqrt{\mathbf{y}^\top \mathbf{\Theta}^{(-1)}\mathbf{y}} \le \sqrt{\mathbf{y}^\top \mathbf{\Theta}'^{(-1)}\mathbf{y}} \le \sqrt{(\mathbf{y}^{(0)})^\top \mathbf{\Theta}'^{(-1)}\mathbf{y}^{(0)}} + \sum_{k=1}^{\infty}\sqrt{(\mathbf{y}^{(2k)})^\top \mathbf{\Theta}'^{(-1)}\mathbf{y}^{(2k)}}.$$

When $k = 0$, we have

$$\sqrt{(\mathbf{y}^{(0)})^\top \mathbf{\Theta}'^{(-1)}\mathbf{y}^{(0)}} \le 4|\alpha_1|\|\beta_1\|_2.$$

When $k \ge 1$, we have

$$\sqrt{(\mathbf{y}^{(2k)})^\top \mathbf{\Theta}'^{(-1)}\mathbf{y}^{(2k)}} \le \sqrt{2\pi}(2k-1)|\alpha_{2k}|\|\Phi^{2k}(\beta_{2k})\|_2 = \sqrt{2\pi}(2k-1)|\alpha_{2k}|\|\beta_{2k}\|_2^{2l}.$$

Thus,

$$\sqrt{\mathbf{y}^\top \mathbf{\Theta}^{(-1)}\mathbf{y}} \le 2|\alpha_1|\|\beta_1\|_2 + \sum_{l=1}^{\infty}\sqrt{2\pi}(2k-1)|\alpha_{2l}|\|\beta_{2l}\|_2^{2l} = o(n)$$

The bound of $\text{tr}(\mathbf{\Theta})$ is simpler to prove.

**Lemma C.4 (Bound on $\text{tr}(\mathbf{\Theta})$)** *Let $n$ denote as the number of training samples. Then $\text{tr}(\mathbf{\Theta}) \le 2n$.*

*Proof.* We have $\boldsymbol{\Theta} \in \mathbb{R}^{n \times n}$. For each $u, v \in V$, as Lemma C.3 shown that

$$\left[\dot{\boldsymbol{\Sigma}}^{(1)}_{(1)}\left(G_u, G_v\right)\right]_{uv} = \frac{\pi - \arccos\left(\left[\boldsymbol{\Sigma}^{(1)}_{(0)}\left(G_u, G_v\right)\right]_{uv}\right)}{2\pi} \leq \frac{1}{2}$$

and

$$\left[\boldsymbol{\Sigma}^{(1)}_{(1)}\left(G_u, G_v\right)\right]_{uv} = \frac{\pi - \arccos\left(\left[\boldsymbol{\Sigma}^{(1)}_{(0)}\left(G_u, G_v\right)\right]_{uv}\right) + \sqrt{1 - \left[\boldsymbol{\Sigma}^{(1)}_{(0)}\left(G_u, G_v\right)\right]_{uv}^2}}{2\pi} \leq 1,$$

we have

$$\boldsymbol{\Theta}_{uv} \leq 2.$$

Thus,

$$\mathrm{tr}(\boldsymbol{\Theta}) \leq 2n.$$

**Combine** Combining Theorem C.1, Lemma C.3 and Lemma C.4, it is easy to see for a constant $c \in (0, 1)$ :

$$L_{\mathcal{D}(F)} = \mathbb{E}_{(v,y) \sim G}[l(F(G, v), y)] \lesssim O(1/n^c).$$

## D   Detailed ablation studies of NeighGen

In this section, we provide in-depth NeighGen studies to empirically explain its power in the cross-subgraph missing neighbor generation. Specifically, we first show the intermediate results of NeighGen by boiling down the generation process into the missing cross-subgraph link generation by dGen and the missing cross-subgraph neighbor feature generation by fGen. Next, we experimentally verify the necessity of training locally specialized NeighGen. Finally, we provide FL training hyper-parameter study on batch size and local epoch to emphasize the robustness of FedSage+.

### D.1   Intermediate results of dGen and fGen.

In this section, we study the two generative components in NeighGen, *i.e.*, dGen and fGen, to explore their expressiveness in reconstructing missing neighbors. Especially, we analyze the outputs from dGen and fGen separately to explain how NeighGen assists in the missing cross subgraph neighbor generation process.

As described in Section 4, both dGen and fGen are constructed as fully connected neural networks (FNNs) whose depths can be varied according to the target dataset. In principle, due to the expressiveness of FNNs [29], dGen and fGen with even very few layers have the power to approximate complex functions. The node degree and feature distributions, on the other hand, are often highly relevant to the graph structure and less complex in nature. In Fig. 1 and Table 1, we provide intermediate results on how dGen and fGen are able to recover missing neighbor numbers and features, respectively.

**Additional details for dGen.**   Fig. 1 shows the break-down performance of dGen on the MSAcademic dataset with M=3, which clearly shows the effectiveness of dGen in recovering the true number of missing neighbors. Notably, though the original output of dGen is a float number, we simply apply the round function to retrieve the integer number of missing neighbors for reconstruction.

**Additional details for fGen.**   As described in Section 4.1, based on the number of missing neighbors generated by dGen, fGen further generates the feature of missing neighbors, thus recovering the incomplete neighborhoods resulting from the subgraph federated learning scenario. Regarding to our ultimate mission in missing neighbor generation as described in Section 4, *i.e.*, locally modeling the original global graph during graph convolution, we evaluate fGen by comparing the NeighGen generated neihgbors with the neihgbors drawn from original whole graph and the ones from original subgraph. Specifically, we present the $L_2$ distance between the averaged feature distributions of neighborhoods from these three types of graphs to show how the NeighGen generated missing neighbors narrow the gap. For simplicity, we use $N(v)$, $N_i(v)$, and $N_i'(v)$ to represent the first-order neighbors of nodes $v \in V$ drawn from the global graph $G$, the original subgraph $G_i$, and the mended subgraph $G_i'$ respectively. Smaller values indicate the locally drawn neighbors ($N_i(v)$ or $N_i'(v)$) being more similar to the true neighbors from the global graph ($N(v)$). The results in Table 1 clearly show the effectiveness of fGen in recovering the true features of missing neighbors.

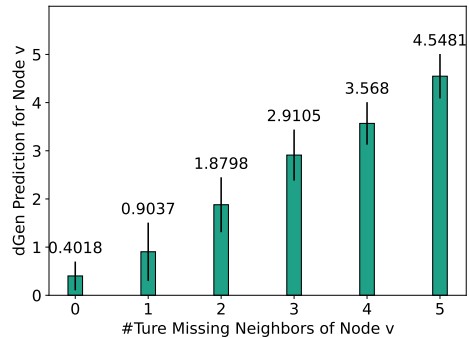

Figure 1: Prediction of dGen for nodes in MSAcademic with $M$=3.

Table 1: Intermediate prediction evaluation for fGen.

| M=3 | Cora | CiteSeer | PubMed | MSAcademic |
|---|---|---|---|---|
| $L_2(N_i'(v), N(v))\pm$ std | **0.0124**±0.0140 | **0.0074**±0.0097 | **0.0034** ±0.0047 | **1.1457** ±1.580 |
| $L_2(N_i(v), N(v))\pm$ std | 0.0168±0.0182 | 0.0101±0.0131 | 0.0046 ±0.0053 | 1.8690±1.8387 |
| **M=5** | **Cora** | **CiteSeer** | **PubMed** | **MSAcademic** |
| $L_2(N_i'(v), N(v))\pm$ std | **0.0262**±0.0885 | **0.0065**±0.0083 | **0.0040**±0.0054 | **1.1245** ±1.5801 |
| $L_2(N_i(v), N(v))\pm$ std | 0.0309 ±0.0897 | 0.0083±0.0115 | 0.0053±0.0060 | 1.8806±1.9695 |
| **M=10** | **Cora** | **CiteSeer** | **PubMed** | **MSAcademic** |
| $L_2(N_i'(v), N(v))\pm$ std | **0.0636**±0.2100 | **0.1569**±0.3310 | **0.0056**±0.0170 | **2.7136** ± 4.5595 |
| $L_2(N_i(v), N(v))\pm$ std | 0.0687±0.2093 | 0.1586 ±0.3307 | 0.0065±0.0171 | 3.2985±4.5686 |

## D.2 Usage of local specialized NeighGens

To empirically explain why we need separate NeighGen functions, we contrast the downstream task performances between FedSage with a globally shared NeighGen, *i.e.*, FedSage with NeighGen obtained with FedAvg, and FedSage with FL obtained local specialized NeighGens, *i.e.*, FedSage+. We conduct ablation experiments on four datasets with $M$=3, and the results are in Table 2. The results clearly assert our explanation in Section 4.3, *i.e.*, directly averaging NeighGen weights across the system degenerates the downstream task performance, which indicates the insufficiency of FedAvg in assisting local data owners in the diverse neighbor generation.

Table 2: Contrast results in node classification accuracy under $M$=3

| Model | Cora | CiteSeer | PubMed | MSAcademic |
|---|---|---|---|---|
| FedSage | 0.8656 | 0.7393 | 0.8708 | 0.9327 |
| (without NeighGen) | (±0.0064) | (±0.0034) | (±0.0014) | (±0.0005) |
| FedSage | 0.8619 | 0.7326 | 0.8721 | 0.9210 |
| with globally shared NeighGen | (±0.0034) | (±0.0055) | (±0.0012) | (±0.0016) |
| FedSage+ | **0.8686** | **0.7454** | **0.8775** | **0.9414** |
| (with local specialized NeighGens) | (±0.0054) | (±0.0038) | (±0.0012) | (±0.0006) |

## D.3 Experiments on Local Epoch and Batch Size

For the proposed FedSage and FedSage+, we further explore the association between the outcome classifiers' performances and different training hyper-parameters, *i.e.*, batch size and local epoch number, which are often concerned in federated learning frameworks.

The experiments are conducted on the PubMed dataset with $M = 5$. To control the variance, we fix the model parameters' updating times. Specifically, for subgraph FL methods, *i.e.*, FedSage and FedSage+, we fix the communication round as 50, while for the centralized learning method, *i.e.*, GlobSage, we train the model for 50 epochs. Under different scenarios, we train the GlobSage model with all utilized training samples in $M$ data owners. Test accuracy indicates how models perform on the same set of global test samples. Results are shown in Table 3 and 4. Every result is presented as Mean (± Std Deviation).

Table 3: Node classification accuracy under different batch sizes with local epoch number as 1.

| Batch Size | FedSage | FedSage+ | GlobSage |
|---|---|---|---|
| 1 | 0.8682($\pm$0.0012) | **0.8782**($\pm$0.0012) | 0.8751($\pm$0.001) |
| 16 | 0.8733($\pm$0.0018) | **0.8814**($\pm$0.0023) | 0.8736($\pm$0.0013) |
| 64 | 0.8696($\pm$0.0035) | 0.8755($\pm$0.0047) | **0.8776**($\pm$0.0011) |

Table 4: Node classification accuracy under different local epoch numbers with batch size as 64. Note that GlobSage is trained with 50 epochs.

| Local Epoch | FedSage | FedSage+ | GlobSage |
|---|---|---|---|
| 1 | 0.8696($\pm$0.0035) | 0.8755($\pm$0.0047) | |
| 3 | 0.8663($\pm$0.0003) | 0.8740($\pm$0.0015) | **0.8776**($\pm$0.0011) |
| 5 | 0.8591($\pm$0.0012) | 0.8740($\pm$0.0011) | |

Table 3 and 4 both evidence the reliable, repeatable therapeutic effects that FedSage+ consistently further elevates FedSage in the global node classification task. Notably, in Table 3, when batch sizes are as small as 16 and 1, FedSage+ accomplishes even higher classification results compared to the centralized model GlobSage due to the employment of NeighGen.

Table 3 reveals the graph learning model can be affected by different batch sizes. As GlobSage is trained on a whole global graph, rather than a set of subgraphs, compared to FedSage and FedSage+, it suits a larger batch size, *i.e.*, 64, than 1 or 16. Both FedSage and FedSage+, where every data owner samples on a limited subgraph, fit better in batch sizes 16. Remarkably, when the batch size equals 1, FedSage is prone to overfit to local biased distribution, while FedSage+ resists the overfitting problem under the NeighGen's assistance, *i.e.*, generating cross-subgraph missing neighbors.

Table 4 provides the relation between the local epoch number and the downstream task performance. For FedSage, more local epochs degenerate the outcome model with more biased aggregated local weights, while FedSage+ maintains a relatively more stable performance in the downstream task. Table 4 empirically evidences that the missing neighbor generator in FedSage+ provides further generalization and robustness in resisting rapid accuracy loss brought by higher local epochs.

Similar to results in Table 2, Section 5, FedSage and FedSage+ exhibit competitive performance even compared to the centralized model. Findings in Table 3 and 4 further contribute to a better understanding of the robustness in FedSage+ compared to vanilla FedSage.