# OpenReview forum: "Subgraph Federated Learning with Missing Neighbor Generation"
_NeurIPS.cc/2021/Conference — NeurIPS 2021 Spotlight_

### Official Review · Reviewer_ehQP · 2021-06-29

**Rating:** 6
**Confidence:** 3

**Summary:**

This paper proposes a federated learning framework for training federated GNNs under the scenario where each dataholder holds disjoint a disjoint subgraph. The paper introduces FedSAGE, which extends GraphSAGE to the federated setting, and FedSAGE+, which further enhances FedSAGE with a missing neighbor generator. The key insight of FedSAGE+ is that the links between subgraphs will be missing, and the neighbor generator is trained to generate missing neighbors. The authors demonstrate their framework using experiments on Cora, Citeseer, Pubmed and MSAcademic, where FedSAGE+ shows improvement over LocalSAGE without the missing neighbor generation.

**Ethical Concerns:**

No.

**Limitations And Societal Impact:**

Limitations are stated.

**Main Review:**

**Strengths**:
- **Novel setting**. I think this paper considers a novel setting of training GNNs in a federated manner, i.e. each party holds a subgraph, and different subgraphs are disjoint, which motivates the problem of missing links. This should be a novel problem setting to consider the topic.
- **Clarity**: This paper is in general well written and easy to follow.

**Weaknesses**:
- **Unclear how to deal with missing edges.** I appreciate the authors' setting of missing neighbors. However, throughout the paper, I can only see that the authors try to reconstruct the missing nodes $V_i^h$, but not the missing edges $E_i^h$ through optimization. Therefore, I am unclear about how the generated pseudo nodes $\tilde{X}_i$ are connected to the remaining of the graph $\hat{G}_i$. Therefore, I cannot clearly see how the missing neighbor generation is done and contributes to the overall model learning.
- **Privacy Concerns**. In Eqn. 7, one term of the optimization objective involves $\sum_{j\in [M]\backslash i}\min_{v_q\in V_j^h}\|x_v^p-x^q\|_2^2$. From my point of view, optimizing this requires the knowledge of $x^q$, which are features from other subgraphs. Does that leak privacy (node features)? From my point of view, it does. Please point out if I have any misunderstandings.
- **Lack of appropriate model analysis**. Please justify the need to use separate NeighGen functions, as no ablation studies on it is done. Also, no ablation studies on parameters are done. I particularly want to see the ablation studies on $\alpha$ in Eqn. 7, as this parameter is related to the use of information from other subgraphs. Also, I want to see the ablation studies on $h$ to see the potential limitations of this model.

**Minor Comments**:
- I was wondering how you manage to train GraphSAGE on Cora and Pubmed to >90\%. From my experience, if you use the split given by DGL, in general, I can only achieve a little >80\%. Please justify the need to use another split.
- Figure 3b, I cannot see the point to show the global distribution of labels, as even if you have the same predicted label distribution, the performance may differ a lot.
- The formulation in Eqn. 1 seems unnecessarily complex. I find it hard to understand, and the notations are not used afterwards.
- Plesse consider using larger datasets, such as Reddit, OGB-products. As Cora, Citeseer, Pubmed are somehow small.

To summarize, due to the limitations in modeling edge information, and the possible leakage of private features, I cannot recommend acceptance at this time.

===================UPDATE============================

The authors successfully address my confusions and concerns through the discussion. I am willing to raise the score from 4 to 6.

**Time Spent Reviewing:**

4

---

> ### Author Response · Authors · 2021-08-10
> **Initial response to Reviewer ehQP**
>
> We thank the reviewer for the time and effort in the review, which acknowledges our contributions on the novel setting and clarity, and provides valuable opportunities for further improvements. Our detailed responses are as follows.
>
> ## 1. Dealing with missing edges.
> When we said “dealing with missing edges”, we never wanted to really reconstruct the missing edges. The whole purpose of NeighGen, as formulated in Eq.(1), is to recover the query distribution of the global graph, which boils down to the recovery of nodes’ neighborhoods. Missing neighbors is indeed a result of missing links, but the recovery of neighbors does not require the recovery of links. In particular, there is no need to actually connect the generated virtual neighbors $\tilde{X}$ back to the remaining graph $\hat{G}$ or original graph $G$, as long as they are properly modeled together with the real neighbors by the GNN classifiers during neighborhood convolution. We thank the reviewer for pointing this out. We are sorry for any possible confusion about this created in the paper, and will carefully fix them in a revision.
>
> ## 2. Possible privacy leakage.
> In Eq.(7), the first term on the RHS is computed fully locally, while the second term is computed on all other clients by (1) sending the predicted node features $\widetilde{x}^p_v$ to all other clients, (2) letting them compute that term and the gradients, and (3) collecting the computed gradients. Therefore, there is no direct data sharing, and only the predicted node features and computed gradients are transmitted among clients. We admit that there could be possible privacy leakage in sending the predicted node features $\widetilde{x}_{v}^{p}$ although it is not direct data sharing, and are aware of possible solutions such as norm clipping and noise injection, which we leave as a future direction. We will make further clarifications on this in a revision.
>
> ## 3. Detailed ablation studies.
> ### 3.1 Usage of separate NeighGen functions.
> We conduct additional ablation experiments on four datasets with M=3 for FedSage with one shared NeighGen obtained from FedAvg versus FedSage+ to empirically explain why we need separate NeighGen functions. The results clearly assert our explanation in Section 4.3 l.213 “cooperation through directly averaging weights of NeighGen across the system can negatively affect its performance, i.e., averaging the weights of a single NeighGen model does not really allow it to generate diverse neighbors from different subgraphs.”
>
> |             Model              |      Cora  | CiteSeer  | PubMed   | MSAcdemic  |
> | ----------------------------- | ------------ | ------------ | ------------- | ----------------- |
> | FedSage with one shared NeighGen   |   0.8333±0.0034   | 0.7257±0.0026    |   0.8691±0.0006   |     0.9210±0.0016         |
> | FedSage+ (with separated NeighGens)   |  **0.8665**±0.0044   |   **0.8526**±0.0007    | **0.8765**±0.0009     |     **0.9414**±0.0006       |
>
> ### 3.2 Hyper-parameter studies on $\alpha$.
> All experiments are implemented with M=3 and h=15%. In the following table, we can observe that choosing a proper $\alpha$, which brings the information from other subgraphs in the system, can constantly elevate the final testing accuracy. Across different datasets, the optimal $\alpha$ is constantly around 1, and the performance is not influenced much unless $\alpha$ is set to very large values like 10.
>
> |          |               | Cora               |                 |               | CiteSeer          |               |
> |----------|---------------|--------------------|-----------------|---------------|-------------------|---------------|
> | Model    | alpha=0.1     | alpha=1            | alpha=10        | alpha=0.1     | alpha=1           | alpha=10      |
> | FedSage+ | 0.8527±0.0021 | **0.8665**±0.0044  | 0.8394±0.0058   | 0.8504±0.0013 | **0.8526**±0.0007 | 0.7198±0.0032 |
> | FedSage  |               | 0.8611±0.0045     |                 |               | 0.8499±0.0013    |               |
>
> |          |               | PubMed             |                 |               | MSAcademic        |               |
> |----------|---------------|--------------------|-----------------|---------------|-------------------|---------------|
> | Model    | alpha=0.1     | alpha=1            | alpha=10        | alpha=0.1     | alpha=1           | alpha=10      |
> | FedSage+ | 0.8514±0.0003 | **0.8756**± 0.0010 | 0.87012±0.0011 | 0.9324±0.0010 | **0.9414**±0.0006 | 0.9261±0.0021 |
> | FedSage  |               | 0.8238±0.0004      |                 |               | 0.9327±0.0005     |               |
>
> ### 3.3 Hyper-parameter studies on $h$.
> All experiments are implemented with M=3 alpha=1. We can observe from the following table that either a too small (1%) or a too large (30%) hiding portion can degrade the learning process, as too little $h$ can not provide sufficient data for training NeighGen, while too large $h$ can result in very sparse local subgraphs for the effective training of GraphSage.
> Referring back to the graph statistics in Table 1 in the paper, the portion of actual missing edges compared to the global graph is within the range of [3.4%, 27.8%], which explains why a value like 15% can mostly boost the performance of FedSage+.
>
> |          |                | Cora               |               |                | CiteSeer           |                |
> |----------|----------------|--------------------|---------------|----------------|--------------------|----------------|
> | Model    | h=1%           | h=15%              | h=30%         | h=1%           | h=15%              | h=30%          |
> | FedSage+ | 0.8527±0.0021  | **0.8665**±0.0044 | 0.8394±0.0058 | 0.8216±0.0029  | **0.8526**±0.0010 | 0.8454±0.0034 |
> | FedSage  |                | 0.8611±0.0045     |               |                | 0.8499 ±0.0013     |                |
>
> |          |                | PubMed             |               |                | MSAcademic         |                |
> |----------|---------------|--------------------|-----------------|---------------|-------------------|---------------|
> | Model    | h=1%           | h=15%              | h=30%         | h=1%           | h=15%              | h=30%          |
> | FedSage+ | 0.8694±0.0012 | **0.8756**±0.0010  | 0.8726±0.0006 | 0.9211±0.0014 | 0.9414±0.0006      | 0.9307±0.0018 |
> | FedSage  |                | 0.8238±0.0004      |               |                | 0.9327±0.0005      |                |
>
> ## 4. Other minor issues.
> * We choose a different training-validation-testing ratio as 60%-20%-20% due to the unique subgraph FL setting, where further splitting the small standard testing dataset of 10% in DGL can result in few testing nodes in some clients,  leading to large testing variances, especially because the simulated splitting of nodes into clients through Louvain algorithm is imbalanced.
> * In Figure 3, we visualize the label distributions of ground-truth and model predictions, to provide possible insight into the reason why FedSage+ can significantly outperform local models. We are aware that there is no definite correlation between label distribution and performance, and we had no intention in claiming for it either.
> * We agree that Eq.(1) is unnecessarily complex and will simplify it in a revision.
> * The PubMed and MSAcademic datasets we use are on the larger side (with 44K and 81K nodes). We will conduct more experiments on larger datasets in the future.

---

> > ### Comment · Reviewer_ehQP · 2021-08-19
> > **Thanks for the reply. Yet I still have unaddressed concerns.**
> >
> > Thanks to the authors for their reply. I now have a clearer picture on the capability of NeighGen through the ablation. However, my following concerns remain.
> >
> >
> > - **Missing edges**. I cannot agree with your point that we needn't actually connect the virtual neighbors back to the original graph. Graph neural networks are powerful because they aggregate neighbors according to edges. Thus, individual nodes (even the virtual ones) must be connected to existing nodes to help the learning of existing nodes.
> >
> >     For example, suppose we have a local subgraph $V = (v_1, v_2, v_3)$, and NeighGen tells me that there are two missing neighbors $(v'_1, v'_2)$ with possible features $x'_1, x'_2$. Now, how to involve $v'_1, v'_2$ in computing the embeddings $v_1, v_2, v_3$? I think this involves connecting $v'_1, v'_2$ to $v_1, v_2, v_3$, and the authors fail to state how to do so.
> >
> > - **Privacy issue**. The authors' reply clears up some of my concerns, but I do have additional questions. Note that the gradient $\nabla_{x_q}\|x_p - x_q\|^2 = 2(x_q-x_p)$, and if a client has the output virtual features ($x_q$), and has the gradient received, can the original data (i.e. $x_p$) from other parties be somehow recovered?
> >
> >     Although in general, this is not possible (because there are many clients, and the gradients $2(x_p-x_q)$ will be summed up). However, in node classification problems, the features are commonly bag-of-word or one-hot vectors, which may facilitate such an attack. The authors may want to justify this issue.

---

> > > ### Author Response · Authors · 2021-08-19
> > > **Thanks and responses for the follow-up questions.**
> > >
> > > We thank the reviewer for reading our responses and adding the insightful questions.
> > >
> > > **Missing edges**. Sorry for the confusion. We did not say *we needn't actually connect the virtual neighbors back to the original graph*. What we mean here is we will generate and connect the virtual neighbors *only* to the corresponding node that we (NeighGen) think has missing neighbors. In your example of $V=(v_1, v_2, v_3)$, NeighGen will be computed on $v_1$, $v_2$ and $v_3$ individually, and predict $\bar{n}_1$, $\bar{n}_2$, and $\bar{n}_3$ as the numbers of the missing neighbors  ($\geq 0$) for $v_1$, $v_2$ and $v_3$, as well as $\bar{X}_1$, $\bar{X}_2$ and $\bar{X}_3$ as the features of the missing neighbors for $v_1$, $v_2$ and $v_3$, as described in the paragraph of l.181. These virtual neighbors are directly connected back to $v_1$, $v_2$, and $v_3$, respectively, which allows the GNN to aggregate the neighborhood information *as if* no neighbors are missing.
> > >
> > > However, what we did not do is further connecting these virtual neighbors to other nodes in the graph, which might be beneficial, because they might act as *bridges* to help the long-range propagation of messages in the graph. We do not know how to do this yet and whether such complication would really bring more benefit, but we are glad to include some discussions about it in a revision, and further study it in future work.
> > >
> > > **Privacy issue**. This is a very interesting question and we are glad that the reviewer has pointed it out. Firstly, in many real-world node classification tasks, the node features are not simply one-hot or bag-of-word (e.g., social networks, healthcare networks, and even publication networks since we have word embeddings nowadays). Secondly, even if the node features are simple/sparse so the individual features can be somehow recovered from the summed gradients, it is hard to guess which individual features are from which clients, which reduces the threat of the attack. Finally, as we discussed in the response, in highly sensitive scenarios, we can apply additional norm clippings and noise injections to the gradients to further increase the difficulty of recovering the individual features.
> > >
> > > In this work, we did not discuss privacy in detail, but rather focused on the basic model design towards utility. We are glad to add some discussions about privacy in a revision, and further study it in future work.
> > >
> > > We thank the reviewer again and please feel free to let us know of any remaining concern(s).

---

> > > > ### Comment · Reviewer_ehQP · 2021-08-20
> > > > **Thanks for the reply. The reply was helpful.**
> > > >
> > > > Having read the reply by the authors, I feel that my concerns can be addressed. I now understand that NeighGen is trained node-wise, rather than subgraph-wise, which clears my confusion on the missing edge issue.
> > > >
> > > > Given that the authors address my confusion, I am willing to raise the score from 4 to 6.

---

### Official Review · Reviewer_VByp · 2021-07-15

**Rating:** 8
**Confidence:** 5

**Summary:**

This paper investigates the problem of subgraph federated learning, with the goal of obtaining a generalized node classification model in a distributed subgraph system without data sharing. To handle this issue, the authors propose a novel framework named FedSage, with interweaving the GraphSage and FedAvg. Besides, a novel missing neighbor generator NeighGen is proposed to tackle the realistic yet unexplored issue of missing cross-subgraph links. Extensive experiments demonstrate the effective performance of FedSage.

**Limitations And Societal Impact:**

Please see Main Review.

**Main Review:**

Originality: Conceptually, this paper studies a novel and important setting of graph federated learning, where smaller subgraphs of a larger global graph are separately owned by different local data owners. The setting is well-motivated with an example of healthcare network distributed in individual hospitals (although no real experiments are done in this setting). Technically, this paper then focuses on an interesting and realistic challenge that the links across subgraphs are missing / not collected by any data owner. This challenge is indeed unique for federated learning on graph data with this subgraph setting, and the authors developed novel techniques to address it. Empirically, the authors conducted experiments on four real-world network datasets from the smaller commonly benchmarked ones and some larger ones. The results show consistent performance improvements of the proposed FedSage+ algorithm, but why the improvements are larger in some particular settings are not that clear.

Quality: The techniques developed in this paper are novel and rich. The leverage of GraphSage with FedAvg is straightforward, and the joint training with the NeighGen model for missing link prediction is new. I also find the particular design of NeighGen pretty novel and interesting, which I have not seen in existing works on GNNs, while the joint prediction of numbers of missing links and virtual missing neighbors is reasonable. Section 4.3 is clear but required some reading to really understand, and the sharing of a NeighGen model is probably conceptually related to meta learning, which is not made clear by the authors. The experiments are comprehensive and mostly convincing, although more detailed analysis could be done with the efficiency (I doubt the algorithm with a shared NeighGen, although without many missing links to predict, can still encounter significant communication overhead, which makes Figure 4(c) not perfectly convincing).

Clarity: This paper is overall very well organized, although the detailed writings are occasionally hard to understand, maybe due to some unnecessarily lengthy sentences. Some typos can also be fixed through more careful proofreading.

Significance: This paper focuses on a realistic setting of subgraph federated learning, with identifying and addressing the unique challenge of missing cross-subgraph links. The novel NeighGen module provides a good remedy, and might be of broader interest to the GNN community beyond federated learning.

**Time Spent Reviewing:**

5 hours

---

> ### Author Response · Authors · 2021-08-10
> **Initial response to Reviewer VByp**
>
> We thank the reviewer for the accurate summarization of our work and valuable suggestions for further improvements.
>
> ## 1. Inconsistent improvements in Table 2.
> We find that FedSage+ exhibits largest performance lifts when the local subgraphs are sparse and more links are missing in the system, such as Cora with M=10 and PubMed. In such situations,  FedSage cannot resolve the information loss brought by the non-trivial amount of missing edges by only aggregating limited local graph information.
>
> ## 2. The conceptual connections to meta-learning.
> Both meta learning and FedSage primarily perceive information from different data sources to obtain a general model. However, meta learning and FedSage are different in their ultimate goals. Specifically, meta learning learns a generalizable model from diverse data sources which can easily adapt to different local tasks with limited fine-tuning, while FedSage aims to learn a generalizable NeighGen model from diverse data owners to help the learning of GraphSage. We thank the reviewer for pointing out this interesting perspective and will add discussions about it in a revision.
>
> ## 3. Runtime results in Figure 4(c).
> We primarily count the classifier training time as we want to compare how the FL obtained classifiers (FedSage and FedSage+) differ from the centralized ones (LocSage, LocSage+ and GlobSage) in time consumption. To provide a more precise study on the runtimes of FedSage+ and LocSage+, we will update this figure by including the training time of NeighGen models and the communication time in a revision.

---

### Official Review · Reviewer_sPrn · 2021-07-20

**Rating:** 6
**Confidence:** 3

**Summary:**

This paper proposed a generalized node classification model over a distributed sub-graph system without data-sharing. The FedSage is a globally shared K-layer graph classifier model, trained with average gradients over distributed sub-graphs without data-sharing. The awareness of potentially existing links between sub-graphs motivate the authors to design a  missing neighbor model to help improve the performance of the node classifier model named FedSage+, which has been empirically shown to outperform the FedSage.



**Main Review:**

NeighGen, consisting of fGen and dGen, is the model proposed to mend the impaired graph and predict the missing neighbors features. fGen is the linear model (parameter multiplying the feature of node i) to predict the missing neighbors of  node i. And dGen tries to predict the missings based on the output of fGen. As they are both simple supervised linear models  (one layer neural network), their effectiveness to  achieve the author’s primary goal - predicting the missing neighbors’ node features, can be questionable. I would expect the author to conduct simple intermediate experiments to show the power of NeighGen on predicting missing neighbors.

Some notations are not defined before use in this section, although I could infer from context. For example, line 183 $\tilde{x}_i$, the number of missing neighbors for node i, Similarly in the eq. (5), $\tilde{n}_v$ is the estimated/predicted number of missing neighbors and $n_v$ is the ground truth number of missing neighbors for node v. Does the $x^q_v$ represent the raw node features for node v in eq(5)?
I also want to know the details about how the dGen model outputs the integer number as it tries to approximate the true missing neighbors numbers.

Since dGen is a supervised model, I would expect the author to present some statistical results to validate that this linear regression dGen model indeed achieves its goal - predicting the number of missing neighbors.

For section 4.3, I am a little confused about the definition of “without data sharing ”.  For example, in the appendix of FedSage + algorithm, on the data owner side, they all send the node features to the Server, will this behavior violate the rule of “no data sharing”. If this is the case, can a Server directly collect all the information from data owner and process , then distribute the results/trained model to all data owners.

In table 2, Cora dataset, the FedSage+ outperforms the GlobalSage, which collects all the information without any missing links. What do you think are possible reasons for this?

Minor Problems and Typos:
Eq.4, the second $\tilde{n}^i$ should be $\tilde{n}_i$
Line 118, to be rigorous, $i \neq j$ can’t be neglected although that’s obvious from context.






**Time Spent Reviewing:**

10 hours

---

> ### Author Response · Authors · 2021-08-10
> **Initial response to Reviewer sPrn**
>
> We thank the reviewer for the time, valuable feedback and suggestions for improvements. The reviewer is in favor of the paper, but has several confusions, which we clarify as follows.
>
> ## 1. Effectiveness of NeighGen (dGen and fGen) and intermediate results.
> As described in l.184, both dGen and fGen are constructed as fully connected neural networks (FNNs), whose depths can be varied according to the target dataset. In principle, due to the expressiveness of FNNs [1], dGen and fGen with even very few layers have the power to approximate complex functions. The node degree and feature distributions, on the other hand, are often highly relevant to the graph structure and less complex in nature. Here, as suggested by the reviewer, we provide intermediate results on how dGen and fGen are able to recover missing neighbor numbers and features, respectively.
>
> The original output of dGen is a float number. To retrieve the integer number of missing neighbors, we simply apply the round function. The following table shows the break-down performance of dGen on the MSAcademic dataset with M=3, which clearly shows the effectiveness of dGen in recovering the true number of missing neighbors.
>
> | # True Missing  | 0        | 1       | 2        | 3        | 4       | 5        |
> |-----------------|----------|---------|----------|----------|---------|----------|
> | dGen Prediction | 0.4018   | 0.9037  | 1.8798   | 2.9105   | 3.5680  | 4.5481   |
> | (std)           | ± 0.2985 | ±0.6049 | ± 0.5696 | ± 0.5294 | ±0.4408 | ± 0.4608 |
>
> Based on the number of missing neighbors generated by dGen, fGen further generates the feature of missing neighbors, thus recovering the incomplete neighborhoods resulting from the subgraph federated learning scenario. In the following table, we show how the generated missing neighbors narrow the gap between the neighborhoods of nodes $v\in V$ drawn from the original subgraph $(q_{G_i,v})$ and the mended subgraph $(q_{G’_i},v})$ compared to the ones retrieved from the global graph $(q_{G,v})$. We present the $L_2$ distance between the feature distributions of the neighborhoods. The results clearly show the effectiveness of fGen in recovering the true features of missing neighbors.
>
> | M=3                         | Cora       | CiteSeer   | PubMed     | MSAcademic |
> |-----------------------------|------------|------------|------------|------------|
> | $L_2(q_{G_i,v}, q_{G,v})$ | **0.0124** | **0.0074** | **0.0034** | **1.1457** |
> | (std)                       | ±0.0140    | ±0.0097    | ±0.0047    | ±1.580     |
> | $L_2(q_{G_i,v}, q_{G,v})$  | 0.0168     | 0.0101     | 0.0046     | 1.8690     |
> | (std)                       | ±0.0182    | ±0.0131    | ±0.0053    | ±1.8387    |
>
> | M=5                         | Cora       | CiteSeer   | PubMed     | MSAcademic |
> |-----------------------------|------------|------------|------------|------------|
> | $L_2(q_{G_i,v}, q_{G,v})$ | **0.0262** | **0.0065** | **0.0040** | **1.1245** |
> | (std)                       | ±0.0885    | ±0.0083    | ±0.0054    | ±1.5801    |
> | $L_2(q_{G_i,v}, q_{G,v})$  | 0.0309     | 0.0083     | 0.0053     | 1.8690     |
> | (std)                       | ±0.0897    | ±0.0115    | ±0.0060    | ±1.8387    |
>
> | M=10                        | Cora       | CiteSeer   | PubMed     | MSAcademic |
> |-----------------------------|------------|------------|------------|------------|
> | $L_2(q_{G_i,v}, q_{G,v})$ | **0.0636** | **0.1569** | **0.0056** | **2.7136** |
> | (std)                       | ±0.2100    | ± 0.3310   | ±0.0170    | ± 4.5595   |
> | $L_2(q_{G_i,v}, q_{G,v})$  | 0.0687     | 0.1586     | 0.0065     | 3.2985     |
> | (std)                       | ±0.2093    | ±0.3307    | ±0.0171    | ±4.5686    |
>
> We thank the reviewer for pointing out such insightful experimental studies and will include these results in our appendix in a revision.
>
> ## 2. Federated training of NeighGen without data sharing.
> In Eq.(7), the first term on the RHS is computed fully locally, while the second term is computed on all other clients by (1) sending the predicted node features $\widetilde{x}^p_v$ to all other clients, (2) letting them compute that term and the gradients, and (3) collecting the computed gradients. Therefore, there is no direct data sharing, and only the predicted node features and computed gradients are transmitted among clients. We admit that there could be possible privacy leakage in sending the predicted node features $\widetilde{x}_{v}^{p}$ although it is not direct data sharing, and are aware of possible solutions such as norm clipping and noise injection, which we leave as a future direction. We will make further clarifications on this in a revision.
>
> ## 3. FedSage+ occasionally outperforms GlobalSage.
> Such results are very occasional, which can be explained by the additional robustness brought by NeighGen. Since NeighGen is trained to recover masked neighbors and further augment the original neighborhoods with generated neighbors, this process can be understood as a self-trained data augmentation, which has the potential to further improve GraphSage, which has no such data augmentation. We will provide discussions about this in a revision and conduct more analysis on it in the future work.
>
> ## 4. Unclear notations and typos.
> * For node $v \in \bar{V}_i$, the predicted number of missing neighbors for $v$ is $\widetilde{n}_v\in\widetilde{n}_i$ and the corresponding predicted neighbors' features are $\widetilde{x}_v\in\widetilde{X}_i$, where $|\widetilde{x}_v|=\widetilde{n}_v$.
> * The reviewer is correct that $x^q_v$ in Eq. (5) means the raw node features of node $q$, where $q$ is a hidden node in $V^h$ and it is a neighbor of node $v$ in the original subgraph.
>
> We will make more careful and thorough checks for unclear notations and typos in a revision.

---

### Decision · Program_Chairs · 2021-09-27

**Decision:**

Accept (Spotlight)

**Comment:**

The paper studies federated learning of graphs. It is assumed that the nodes of the graph are separated between the parties and cross-party edges are not observed. The authors suggest algorithms for learning in this scenario and experiment with them on several datasets.

The reviewers had some concerns about this work, but the authors have used the discussion to explain some ideas. This allowed the reviewers to reach a consensus that this work is above the acceptance threshold.